# Mixed evidence for the rhythmicity of auditory perceptual judgements in humans

**Cécile Fabio, Christoph Kayser\***

Department for Cognitive Neuroscience, Faculty of Biology, Bielefeld University, Bielefeld, Germany

## eLife Assessment

This high-N, multi-task study offers a comprehensive examination of rhythmicity in behavioral performance during listening. It presents a **valuable** set of findings that reveal task- and ear-specific effects, challenging the notion of a universal rhythmicity in auditory perception. The evidence is **solid** and the work is likely to be of significant interest to behavioral and cognitive scientists focused on perception and neural oscillations.

**\*For correspondence:**
christoph.kayser@uni-bielefeld.de

**Competing interest:** The authors declare that no competing interests exist.

**Abstract** Numerous studies advocate for a rhythmic mode of perception. However, the evidence in the context of hearing remains inconsistent. We propose that the divergent conclusions drawn from previous work stem from conceptual and methodological issues. These include ambiguous assumptions regarding the origin of rhythmicity, variations in tasks and attentional demands, and differing analytical approaches for statistical testing. To address these points, we conducted a series of experiments in which human participants performed auditory tasks involving monaural targets presented against binaural white noise backgrounds, while also recording eye movements. These experiments varied in whether stimuli were presented randomly or required motor initialisation, the necessity of memory across trials, and the manipulation of attentional demands. Our findings challenge the notion of universal rhythmicity in hearing, but support the existence of paradigm- and ear-specific fluctuations in sensitivity and biases at multiple frequencies. The rhythmicity for sounds in the left and right ears appears independent among participants, and the rhythmicity in performance is possibly linked to oculomotor activity and attentional requirements. Overall, these results may help to resolve conflicting conclusions drawn in previous work and provide specific avenues for further studies into the rhythmicity of auditory perception.

## Introduction

Many studies over the past decades promote the concept of a 'rhythmic mode' of perception (*Keitel et al., 2022*; *VanRullen, 2016*; *VanRullen and Koch, 2003*). Following this, perception is an active process, shaped not only by external stimuli but also by intrinsic states of the sensory pathways and their interactions with the motor system (*Schroeder et al., 2008*; *Benedetto et al., 2020*; *Morillon et al., 2015*; *Lima et al., 2016*; *De Kock et al., 2021*). This led to the idea of 'perceptual cycles', whereby stimuli occurring during specific moments in time are more likely to be perceived, or are perceived more accurately, compared to those during other moments (*VanRullen, 2016*; *Latour, 1967*; *Pöppel, 1970*). In the context of hearing, this notion of a rhythmic mode of hearing is supported by the prevailing rhythmic structure in neural activity along the auditory pathways, which aligns with the typical multi-scale structure of vocalisations and speech (*Giraud and Poeppel, 2012*; *Ding et al.,*

2016; *Obleser and Kayser, 2019*). Conceptually, it has been proposed that a rhythmic mode may be the default operational mode for listening and that a continuous mode becomes engaged under conditions of prolonged vigilance or high arousal (*Schroeder and Lakatos, 2009*; *Lakatos et al., 2016*; *Milne et al., 2021*). Psychoacoustic studies indeed suggest that our ability to detect or discriminate sounds can vary periodically relative to a reference time point, such as the onset of an experimental trial or another sound (*Ho et al., 2019*; *Ho et al., 2017*; *Hickok et al., 2015*; *Farahbod et al., 2020*; *Zoefel and VanRullen, 2017*). For example, one study showed that pitch judgments for tones presented on a background noise vary at the scale of 6–8 Hz depending on their timing relative to this background (*Ho et al., 2017*). Neuroimaging studies further support this concept by linking large-scale rhythmic activity to perceptual judgements (*Ng et al., 2012*; *Strauß et al., 2015*; *Henry et al., 2016*; *Wöstmann et al., 2019*; *ten Oever and Sack, 2015*; *Florin et al., 2017*), and in vivo recordings confirm the prominence of rhythmic changes in neural excitability and information transmission directly along the auditory pathways (*Lakatos et al., 2005*; *Lakatos et al., 2013*; *Kayser et al., 2015*; *Guo et al., 2017*).

However, it has been difficult to obtain coherent evidence for perceptual cycles in hearing based on purely behavioural data. While some studies reported positive results, others failed to find such or reported only weak evidence for rhythmicity (*Ho et al., 2019*; *Ho et al., 2017*; *Hickok et al., 2015*; *Farahbod et al., 2020*; *Zoefel and VanRullen, 2017*). This mixed evidence may result from multiple aspects of the previous work, as discussed in the following. To address this conundrum, we probed for rhythmicity in auditory perceptual judgments systematically across different tasks and analysis approaches. We here focus on the question of whether perceptual judgements exhibit rhythmicity in the absence of explicitly temporally entraining sounds, which is related but different from the question of whether rhythmicity persists following the presentation of sounds with explicit temporal structure (*Obleser and Kayser, 2019*; *Zoefel and VanRullen, 2017*), similar to studies in the visual domain probing rhythmicity of perception following cued attention (*Fiebelkorn and Kastner, 2019*; *Fiebelkorn et al., 2013*; *Helfrich et al., 2018*).

Previous listening studies, for example, presented brief sounds during unstructured noise and showed that participants' judgements vary with the delay between the onset of the noise and the target. Yet, the reported rhythmicity varies in time scale and significance (*Ho et al., 2019*; *Ho et al., 2017*; *Ng et al., 2012*; *Sun et al., 2022*; *Kayser, 2019*). One reason may be the assumptions about where along the neural pathways' rhythmicity emerges (*Sun et al., 2022*; *İlhan and VanRullen, 2012*; *Zoefel and VanRullen, 2015*; *Haegens and Zion Golumbic, 2018*; *Figure 1A*). Some studies employed binaural targets and hence assumed that rhythmicity is either present at the same frequency in monaural pathways or arises after binaural convergence (*Hickok et al., 2015*; *Farahbod et al., 2020*; *Sun et al., 2022*; *Kayser, 2019*). In contrast, other studies directly tested monaural targets and reported effects at different frequencies for each ear (*Ho et al., 2019*; *Ho et al., 2017*). Such an ear-specificity of rhythmic perceptual sampling is not unreasonable given hemisphere differences in the prominence of rhythmic activity in auditory cortices and their putative link to a functional differentiation of left and right auditory networks (*Gross et al., 2013*; *Keitel et al., 2018*; *Albouy et al., 2020*; *Flinker et al., 2019*). Yet both the robustness of these results and the precise origin of putative rhythmicity in listening remain unclear. In order to probe the ear-specificity of rhythmicity, we presented targets monaurally and analysed the data from individual ears, but also when collapsing across ears (*Figure 2*).

A second reason may be the use of different listening tasks, which range from target detection to target discrimination or the continuous sampling of acoustic signals over time (*Kayser, 2019*). To account for this, we compared two auditory tasks that differ in their requirements to contrast stimuli solely within or between trials. A third reason may result from differences in statistical sensitivity, which can be attributed to different analysis methods used to quantify rhythmicity and variations in the sample size in previous work (*Zoefel et al., 2019*; *Tosato et al., 2022*; *Brookshire, 2022*; *Lundqvist and Wutz, 2022*). To address this, we compared different approaches to quantify rhythmicity in behavioural data. We ensured that these are comparable 'in principle' by calibrating their statistical specificity on simulated data. And in addition to testing the significance of effects within the specific participant sample, we probed whether putative effects generalize across participant samples using bootstrap simulations (*Kulesa et al., 2015*), thereby avoiding the use of a single threshold applied to one specific participant sample to determine the presence or absence of effects.

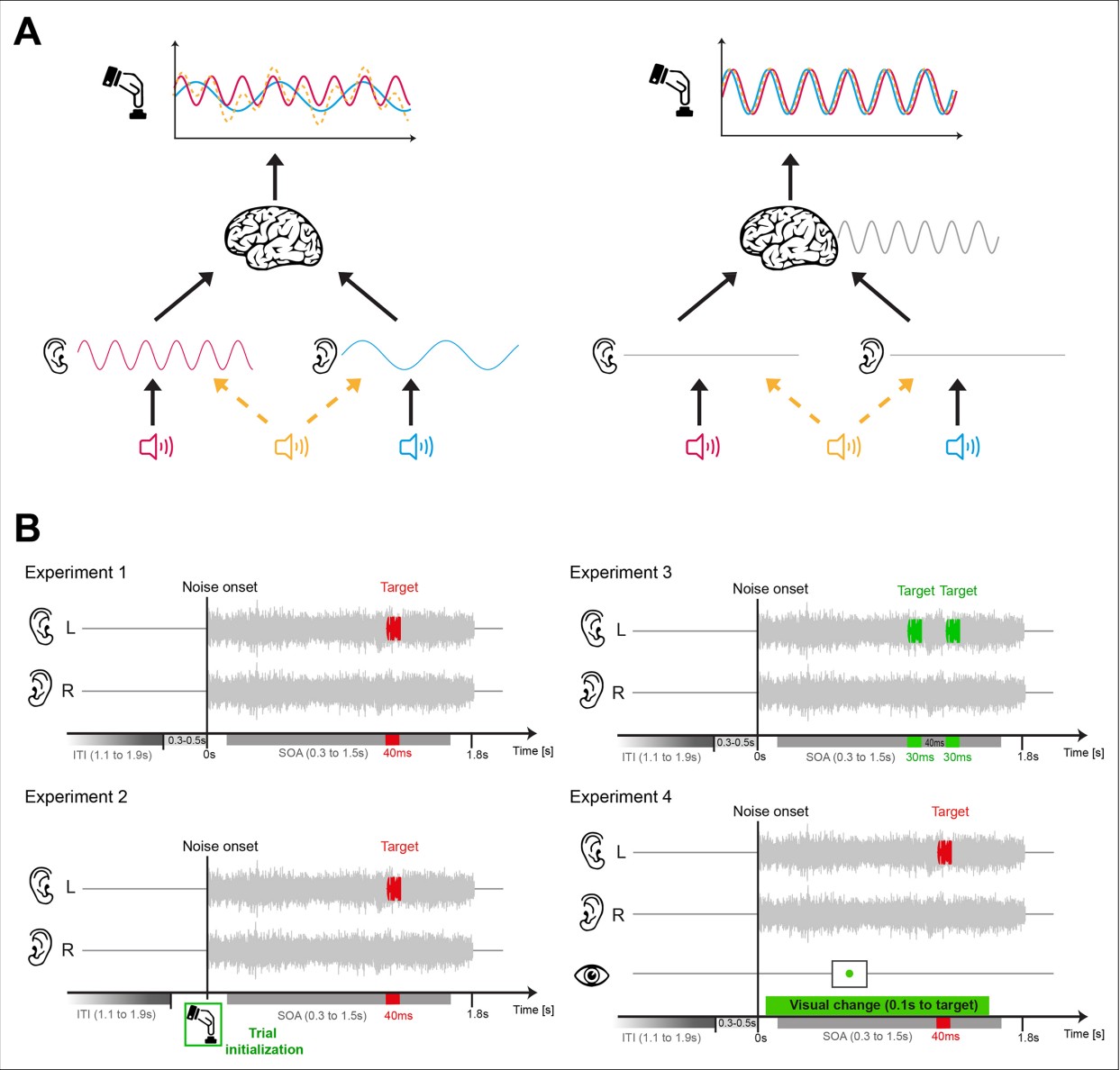

**Figure 1.** Experimental design and rationale. (**A**) Rhythmicity in behavioural data could have several origins. On the one extreme are rhythmic processes in monaural auditory processes that operate at different frequencies for sounds in each ear (left scheme). As a result, one would observe signatures of rhythmicity in behavioural data for monaural targets at different time scales, and a yet different pattern when presenting binaural stimuli (orange). On the other extreme are high-level processes possibility more tied to cognition than sensation, that operate in the same manner on any sound and which give rise to the same rhythmicity in behavioural data following monaural or binaural stimuli (right scheme). (**B**) General design of the four experiments. In each experiment, independent white noise was presented to each ear over a period of 1.8 s after a pseudo-random fixation period (0.3–0.5 s). The task-relevant sounds were presented monaurally and at random time points between 0.3 and 1.5 s following the noise onset and either comprised one (40 ms) or two (30 ms) sounds. Experiments 1, 2, 4 featured a tone discrimination task with a decision criterion fixed across trials (tones categorised as 'low' or 'high'). Experiment 3 featured a within-trial discrimination of two subsequent tones. Experiments 1 and 2 differ in that sound presentation was automatically paced or required manual initialisation by the participant. Experiment 1 and 4 differed in that the latter also required participants to perform a dual task on in the visual fixation dot. This was intended to divert attention across sensory modalities.

A fourth reason may relate to differential requirements for attention or motor-related processes in previous studies. It has been proposed that a rhythmic mode of listening is specifically engaged when attention is divided across the senses, while a strong focus on hearing may shift perception towards a continuous mode (*Schroeder and Lakatos, 2009*; *Schroeder et al., 2010*; *O'Connell et al., 2020*). Based on this, we contrast the same listening task when performed in isolation or in combination with a dual task diverting attention to the visual modality. Changes in attention may also directly

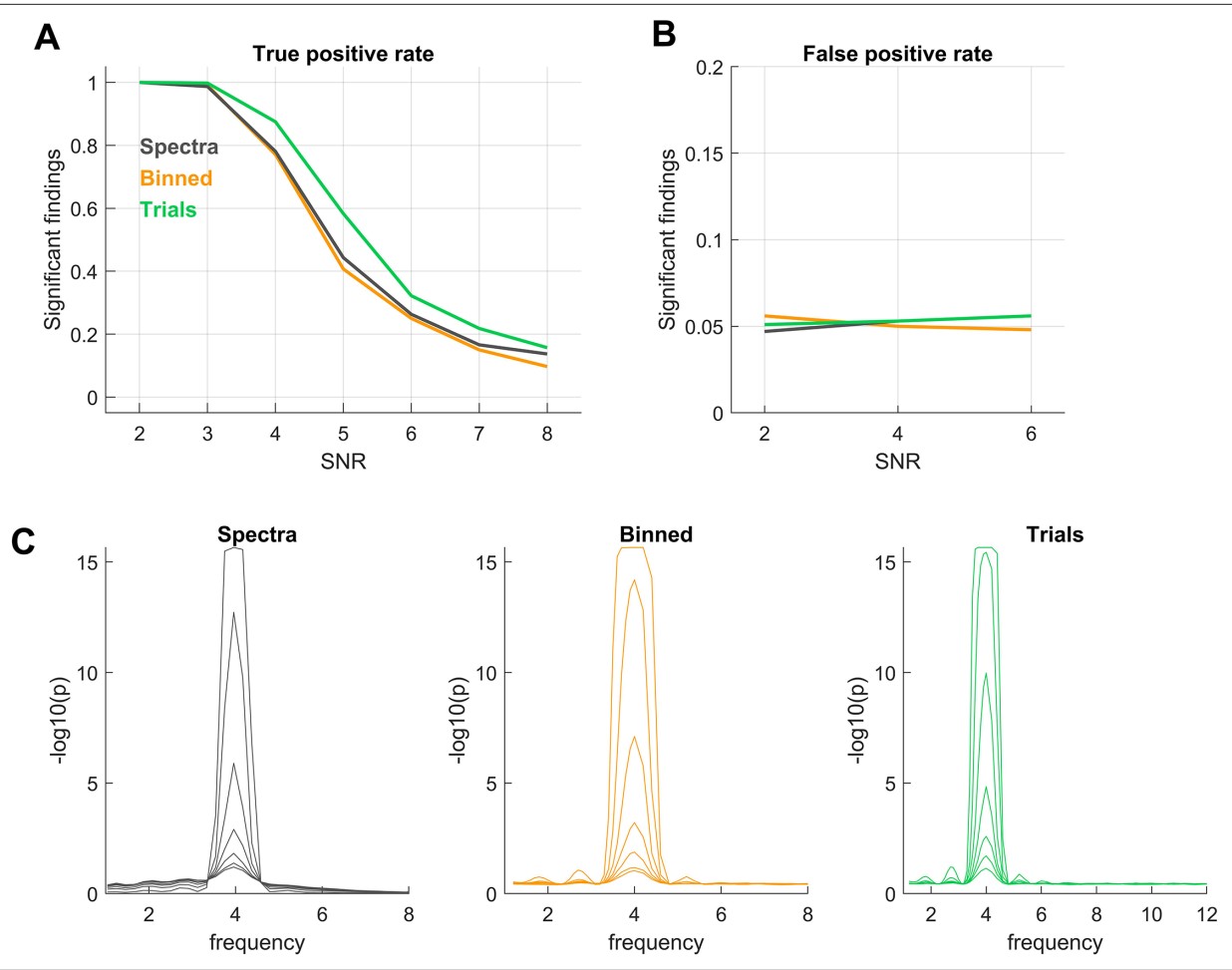

**Figure 2.** Calibration of analysis approaches on simulated data. We simulated data with and without rhythmicity to calibrate the statistical specificity (false positive rate) of each approach. The approaches are based on the spectra derived from delay-binned data ('Spectra'), the vector strength of a rhythmic component in linear models applied to delay-binned data ('Binned') and the vector strength of a rhythmic component in linear models applied to single-trial data ('Trials'). For each, we determined the respective first-level threshold (i.e. method-specific p-value) that results in a false-positive rate of about 0.05 when probing for rhythmicity at any frequency. (**A**) Sensitivity of each approach in detecting a rhythmic effect for data generated with a rhythmic effect and different signal-to-noise ratios (SNR). (**B**) Specificity, shown here as false positive rate in detecting a rhythmic effect in data generated without such an effect (calibrated to about 0.05). (**C**) Illustration of the (log-transformed) first-level p-values for simulated data with an effect at 4 Hz and different SNRs, showing the frequency specificity of each approach in detecting an effect.

reflect as changes in oculomotor behaviour and it is known that the oculomotor system is intricately connected with the auditory pathways (*O'Connell et al., 2020*; *Gehmacher et al., 2022*; *Köhler and Weisz, 2023*; *Gruters et al., 2018*; *Bulkin and Groh, 2012*; *Populin et al., 2004*; *Werner-Reiss et al., 2003*; *Schneider et al., 2014*; *Leszczynski et al., 2023*). More generally, it has been suggested that motor activity provides a scaffold for temporally organising perception, and hence it is conceivable that differences in motor commands contribute to shaping rhythmicity in behavioural data (*Morillon et al., 2015*; *Lima et al., 2016*; *De Kock et al., 2021*; *Morillon et al., 2014*; *Zalta et al., 2020*). For example, paradigms requiring participants to start the presentation of the task-relevant stimulus by an explicit motor action may engage different mechanisms relating to temporal prediction or attention than paradigms presenting stimuli at pseudo-random intervals. Presenting stimuli at a participant's preferred time points may facilitate the attentional engagement and allow participants' brains to fine-tune ongoing processes to process sensory information optimally (*Schroeder et al., 2008*; *Obleser and Kayser, 2019*; *Schroeder and Lakatos, 2009*; *Weisz et al., 2014*; *Herbst et al., 2022*). We tested this by contrasting paradigms requiring, or devoid of, the explicit initialisation of target presentation and by including eye-tracking data about oculomotor activity and pupil size in the analysis.

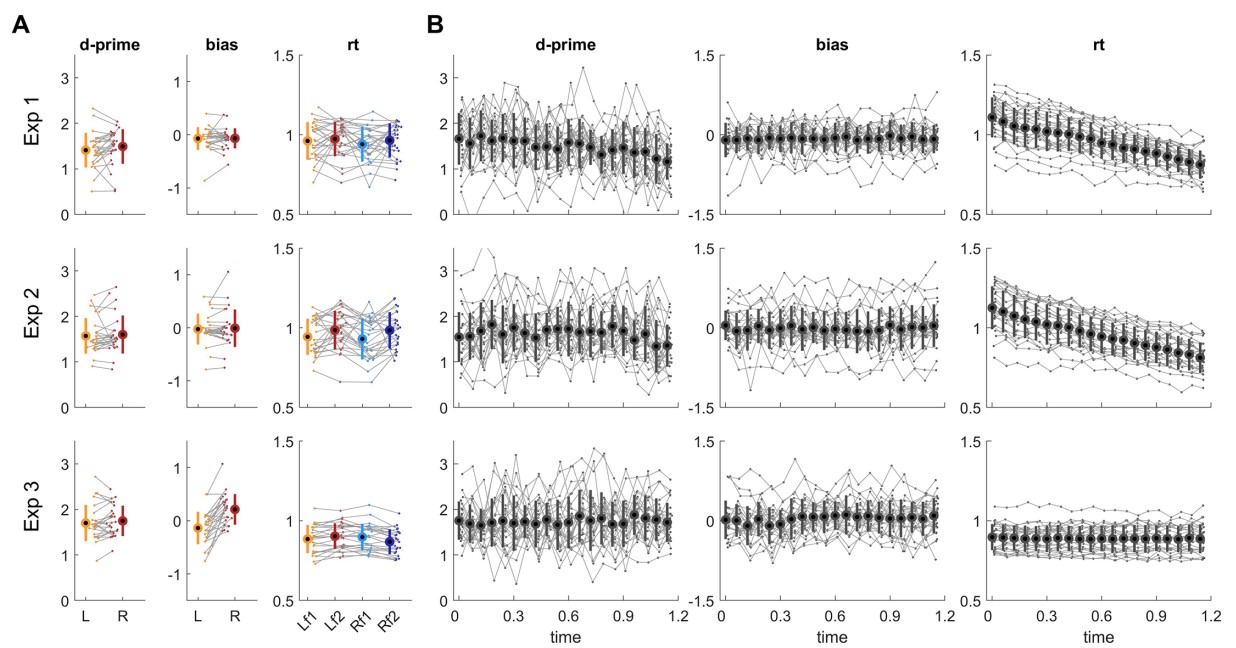

**Figure 3.** Illustration of the data for experiments 1–3. (**A**) Sensitivity (d-prime), bias, and reaction times (rt) regardless of target delay. These metrics are shown separately for each ear (**R, L**) and reaction times (in seconds) are shown separately for each stimulus condition (f1, f2; corresponding to the two target frequencies in experiments 1 and 2 or the two orders of pitch in experiment 3). (**B**) The same metrics for trials with targets presented at specific delays within the 1.2 s range of target uncertainty. Grey dots and lines denote the individual participant data, thick dots and error bars denote the group average and standard deviation.

Motivated by previous work, we capitalised on a monaural pitch discrimination task and probed how different metrics of participants' judgements (sensitivity, bias, and reaction time) vary as a function of the delay between the onset of a background noise and the target tone. We implemented four variations of this paradigm, each testing more than 25 participants (*Figure 1B*). One experiment tested the original paradigm by *Ho et al., 2017*, and featured automatically paced trials asking participants to categorise the pitch of a target tone as 'low' or 'high' (Experiment 1). A variation of this experiment provided participants with explicit knowledge about the timing of individual trials, by requiring them to manually initialise stimulus presentation for each trial (Experiment 2). In a third experiment, we implemented a within-trial pitch discrimination task (Experiment 3), which does not require the implicit comparison of a target stimulus to an absolute reference that is maintained across trials (as is the case in experiment 1). Given that a between-trial discrimination task requires memory of a stimulus boundary across trials, it is conceivable that the presence or absence of such memory and related top-down processes may affect rhythmicity in behaviour. And finally, we combined the auditory task with a dual-task diverting attention to a visual fixation point, in which we also collected eye tracking data (Experiment 4). In the following, we present a number of experiments, different analytical approaches for quantifying rhythmicity and probing for rhythmicity in different behavioural metrics. Given that the individual statistical outcomes (i.e. significances) have to be taken with care, we base our interpretations on the prevalence of effects among random variations in the participant sample both within and between experiments.

## Results

### Illustration of the data for experiments 1-3

The experiments involved auditory discrimination tasks performed on monaural target sounds presented at different delays relative to the onset of a binaural white noise background. *Figure 3* illustrates the main aspects of this data. Panel A shows the overall sensitivity and bias for targets presented to each ear and reaction times for individual ears and targets. Panel B shows the behavioural metrics as a function of the delay using binned data. This illustrates temporal structure in behaviour,

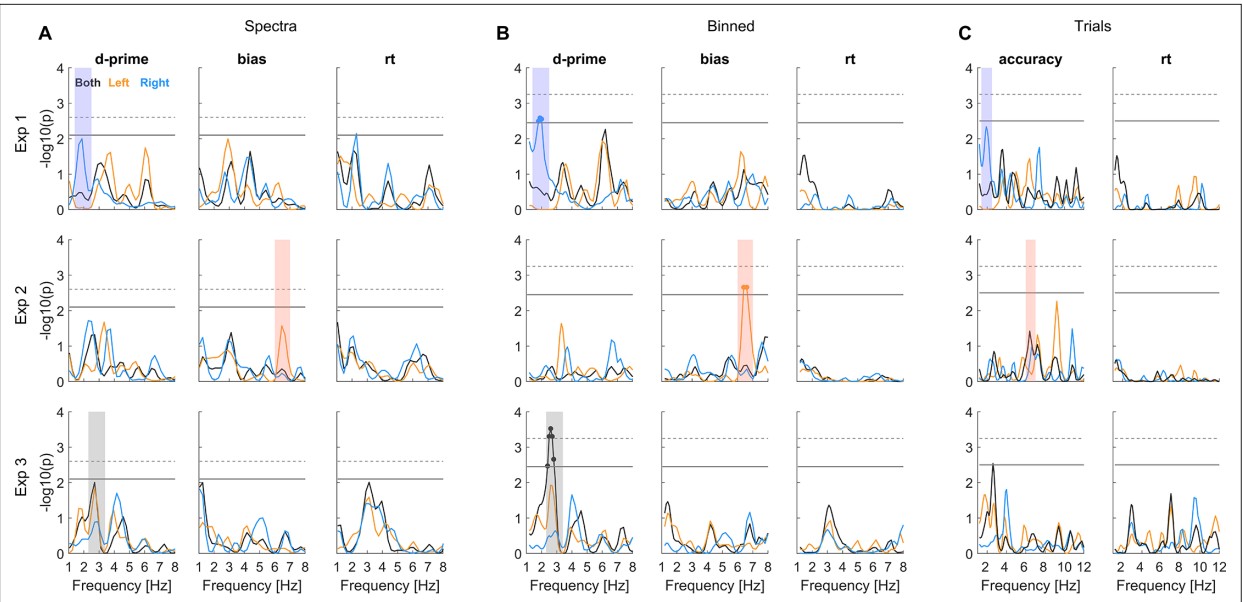

**Figure 4.** Significance of rhythmicity in the behavioural data. The individual panels show the group-level (first-level) p-values for each approach (**A–C**) and experiment together with the statistical cut-off used to determine significance (thick grey line; corresponding to p<0.05 corrected for multiple tests across frequencies, calibrating false-positive rate across analyses). For comparison, the dashed grey line also shows the cut-off at p<0.01. The coloured shadings indicate significant effects to facilitate their comparison across panels. The precise frequencies with significant effects were: Exp 1: Binned/d-prime: 1.8–2.0 Hz; Exp 2: Binned/bias 6.4–6.6 Hz; Exp 3: Binned/d-prime 2.4–2.8 Hz.

such as decreasing sensitivity or reaction times for targets presented late in the trial. This is in line with previous studies reporting temporal structure in behavioural data at multiple time scales, which is often observed in perceptual decision-making paradigms and may reflect individual strategies for analysing the sensory environment, leakage in decision processes, or the urgency to respond (*Kayser, 2019*; *Okazawa et al., 2018*; *Waskom et al., 2018*). Note that this visualisation of the data implicitly assumes that all participants exhibit rhythmicity (if they do) at the same frequency and phase. However, the assumption about the same phase may not be warranted and is not a requirement in the following statistical analyses.

## Statistical evidence for rhythmicity in experiments 1-3

We report evidence for rhythmicity in the behavioural data in two ways: first, as significance of effects in the specific sample of the collected data and then as the prevalence of effects when generalizing over variations of participants drawn from this sample.

*Figure 4* shows the p-values for a rhythmic effect (vs. a suitable null distribution) for each behavioural metric and the three approaches. We determined for each experiment and metric whether there was a significant effect in any of the analysis approaches (at p<0.05, corrected for multiple tests across frequencies; indicated by the grey lines). For experiment 1, this included an effect for d-prime around 2 Hz for the right ear, for experiment 2, an effect for bias around 6.5 Hz for the left ear, and for experiment 3, an effect in d-prime around 3 Hz for the combined-ear data. These effects are highlighted across approaches (shading in *Figure 4*). Remember that experiments 1 and 2 differed in that trials were automatically paced (experiment 1) or self-initialised (experiment 2), while experiments 1 and 3 differed in that the relevant stimulus information had to be compared between trials (experiment 1) or within-trials (experiment 3). (*Figure 5*).

To probe whether and which effects generalise across random variations in the participant sample, we used a resampling approach (*Figure 6*). This revealed that the above-mentioned effects prevail for at least 50% of the simulated experiments, corroborating their robustness within the participant sample. However, the prevalence data also provide critical insights beyond those obtained from the significance in *Figure 4* and suggest that conclusions drawn from this significance-testing approach may be misleading.

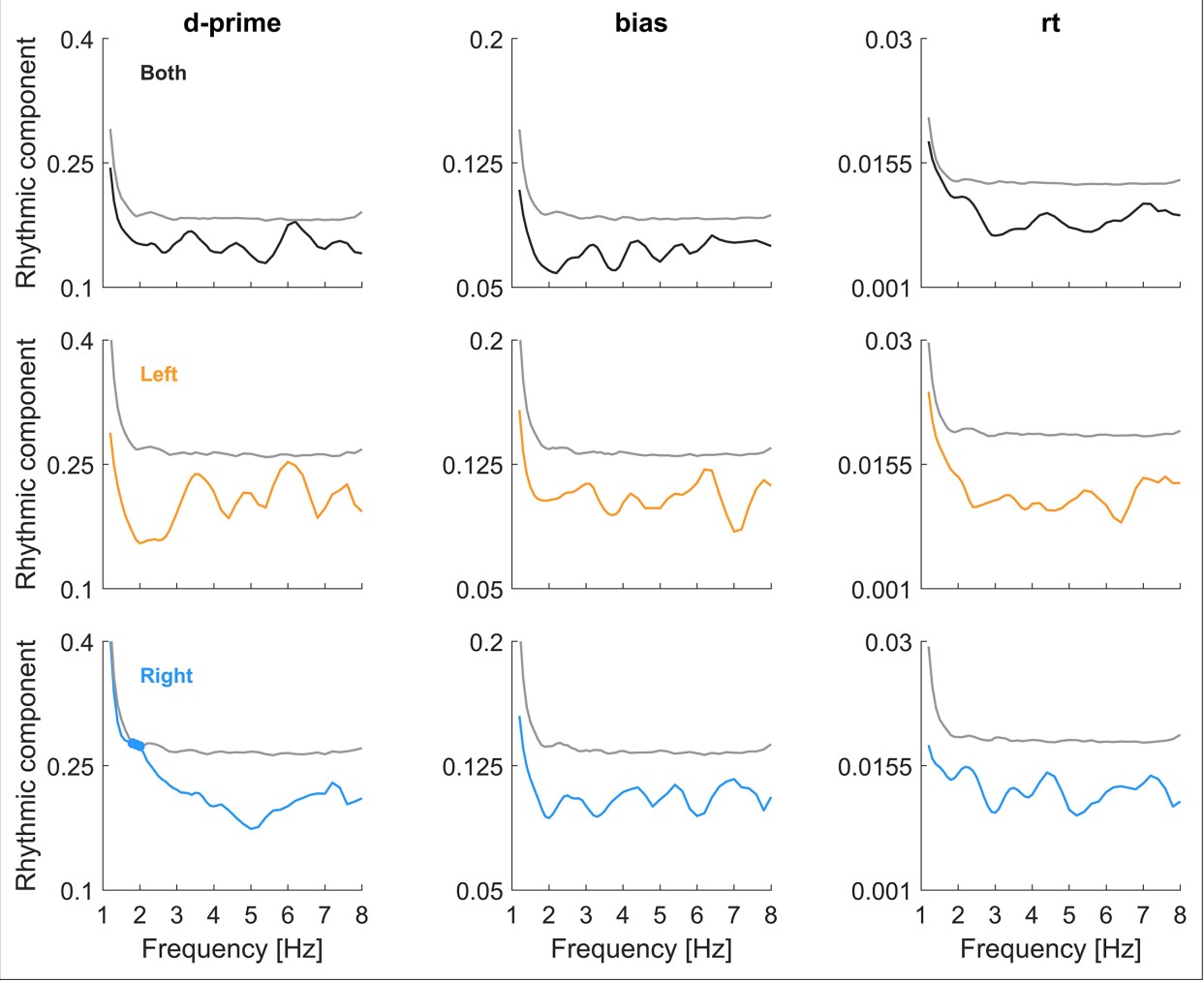

**Figure 5.** Rhythmic effects in the binned analysis for experiment 1 in the actual data (coloured) and the surrogate data. The rhythmic effect here is the combined vector strength of the sine and cosine predictors in the binned analysis, averaged across participants and shown for each metric and ear (colour-coded). For the surrogate data, we picked the percentile corresponding to the significance level of $p=0.05$ and showed the respective group-average surrogate data. In experiment 1, only d-prime for the right ear data was significant around 2 Hz. The figure also illustrates how the effective strength of a rhythmic predictor required to achieve significance varies between metrics, frequencies, and ear combinations.

For experiment 1, sensitivity is modulated in the right ear at around 2 Hz and the left ear around 6.5 Hz with comparable prevalence in multiple approaches (Spectra: 45% vs 38%; Binned: 35% vs 55%; see *Figure 6*). Furthermore, the prevalence of an effect for bias is also considerable (e.g. Spectra 43%). Collectively, this suggests that, in fact both, perceptual sensitivity and bias, may be modulated and in both ears, but at different frequencies. For experiment 2, the effect in bias for the left ear around 6.5 Hz does not seem to be accompanied by a corresponding effect in the right or the combined-ear data at the same frequency (prevalence below 10%). However, there is considerable prevalence of an effect in sensitivity for the right ear around 2 Hz (Spectra 38%), suggesting that also in this experiment, both ears may exhibit rhythmicity, again possibly at distinct frequencies. For experiment 3, all three approaches reveal the prevalence of rhythmicity in sensitivity around 2–3 Hz for the combined-ear data (Spectra 44%, Binned 78%, and Trials 49%) and to a smaller degree also in the left ear (Spectra 40%, Binned 34%) without direct counterpart for the right ear at the same frequency. Finally, especially the spectral approach also suggests a considerable prevalence of effects in reaction times. Hence, based on this analysis of the data, we conclude that multiple behavioural metrics reveal a largely comparable prevalence of rhythmicity across experiments and ears, and that the precise frequencies and prevalence vary between approaches and experiments.

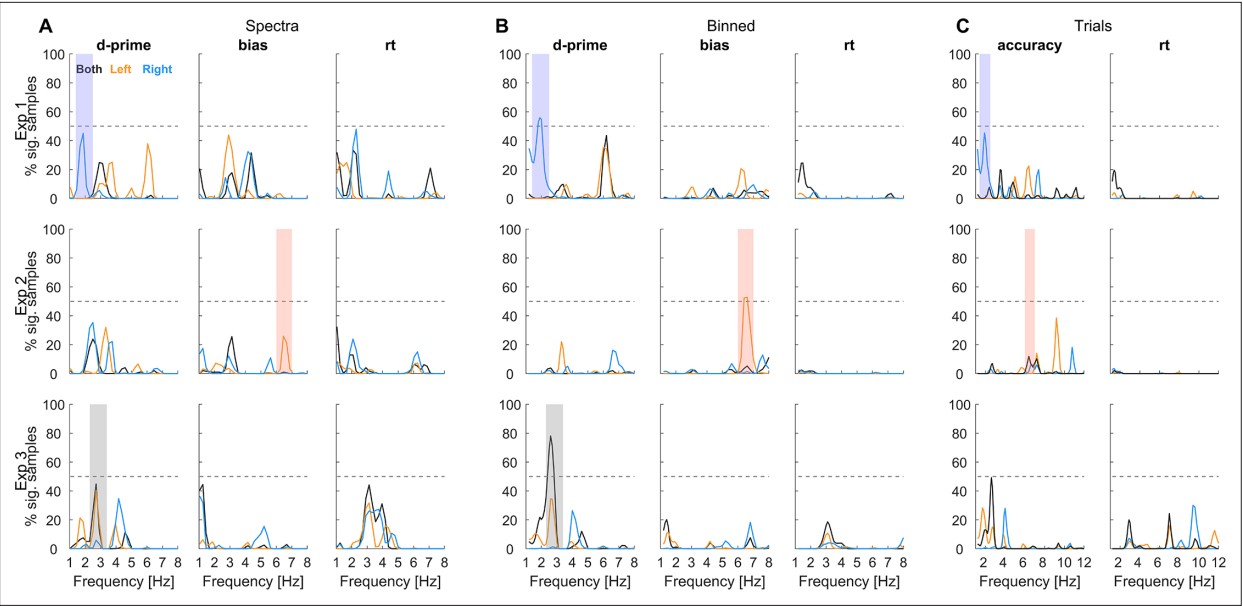

**Figure 6.** Prevalence of significant effects across random variations in the participant sample. Using a resampling approach, we determined the percentage of participant samples that yield a significant effect when repeating the analysis. For each simulated participant sample, we determined significant effects at $Pp<0.05$ (as in *Figure 4*) and then counted the number of simulations with effects at each frequency. A value of above e.g., 50% (dashed line) indicates that when randomly sampling from within the collected participants more than 50% of such simulated experiments yield a significant effect. The coloured shadings highlight the same effects as shown in *Figure 4*.

## Evidence for the ear-specificity of rhythmicity

To determine whether rhythmicity indeed prevails at distinct frequencies for the two ears consistently across experiments, we averaged the prevalence data across experiments 1–3 (*Figure 7*). We restricted this analysis to sensitivity and bias and the two approaches showing the most prevalent effects above (Spectra, Binned).

This summary suggests that the two ears indeed tend to exhibit the highest prevalence of effects at complementary frequencies. This is seen both for sensitivity and bias and in both approaches. In particular, peaks for sensitivity in the right ear prevail around 2 Hz and 4 Hz, and for the left ear around 3 Hz and 6 Hz. A similar picture emerges for bias, though these results differ more between analysis approaches. Given that the prevalence data reflect the likelihood of observing a significant effect for a given random sample of participants, it is possible that for a specific sample, one may observe only one of these spectral peaks. This may be one reason for the diversity of frequencies and effects reported in the literature, given that most studies focused on the effects that exist in a specific and unique participant sample.

The data also leave the possibility that the rhythmic effects in the left and right ears emerge in distinct groups of participants and may not be strictly linked. To probe this, we correlated the

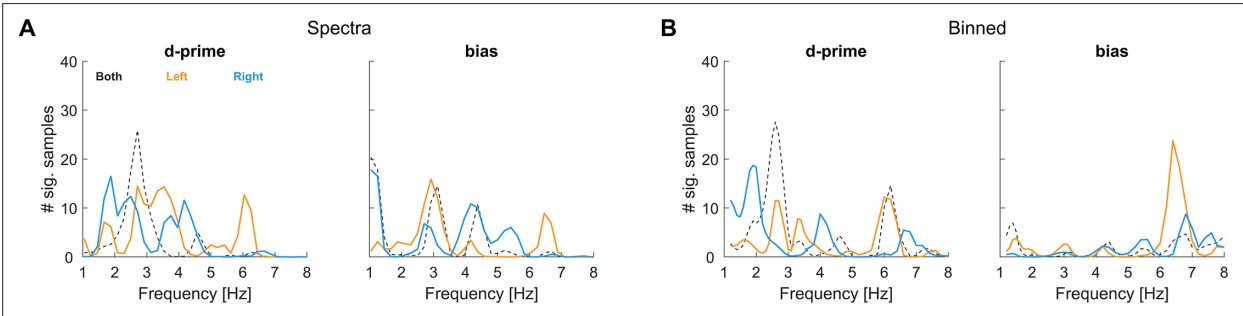

**Figure 7.** Averaged prevalence across experiments 1–3. We focus on the two approaches and metrics yielding the strongest effects in the data for individual experiments. To allow better visibility of the differences between individual ears, the combined-ear data are shown dashed.

occurrence of significant effects in the left and right ears across the 5000 bootstrapped samples. Specifically, we coded the presence of significant effects in the right ear around 1.5–2 Hz and of effects in the left ear around 5.5–6 Hz for the data of experiment 1 as binary variables. The correlations of these co-occurrences of effects in the two ears were minimal (Spectra: $r=-0.01$, $p=0.46$, Binned: $r=0.018$, $p=0.20$, n=5000), suggesting that any rhythmicity for left and right ears may not emerge simultaneously but in part prevails in separate groups of participants.

### Role of arousal and eye mobility

Experiment 4 was designed to test additional properties of putative rhythmicity in behavioural data. First, the experiment contrasted blocks featuring a pure auditory task with blocks including the diversion of attention by a dual visual task. Second, we measured eye movements to probe whether the presence of rhythmicity is linked to arousal (indexed using pupil dilation) or to the overall mobility of the oculomotor system (indexed using fixation stability).

Participants performed the dual task well. Their sensitivity to the intensity changes of the fixation dot was high (d-prime, 3.96±0.12, mean ± s.e.m., n=36) and they exhibited no obvious bias in

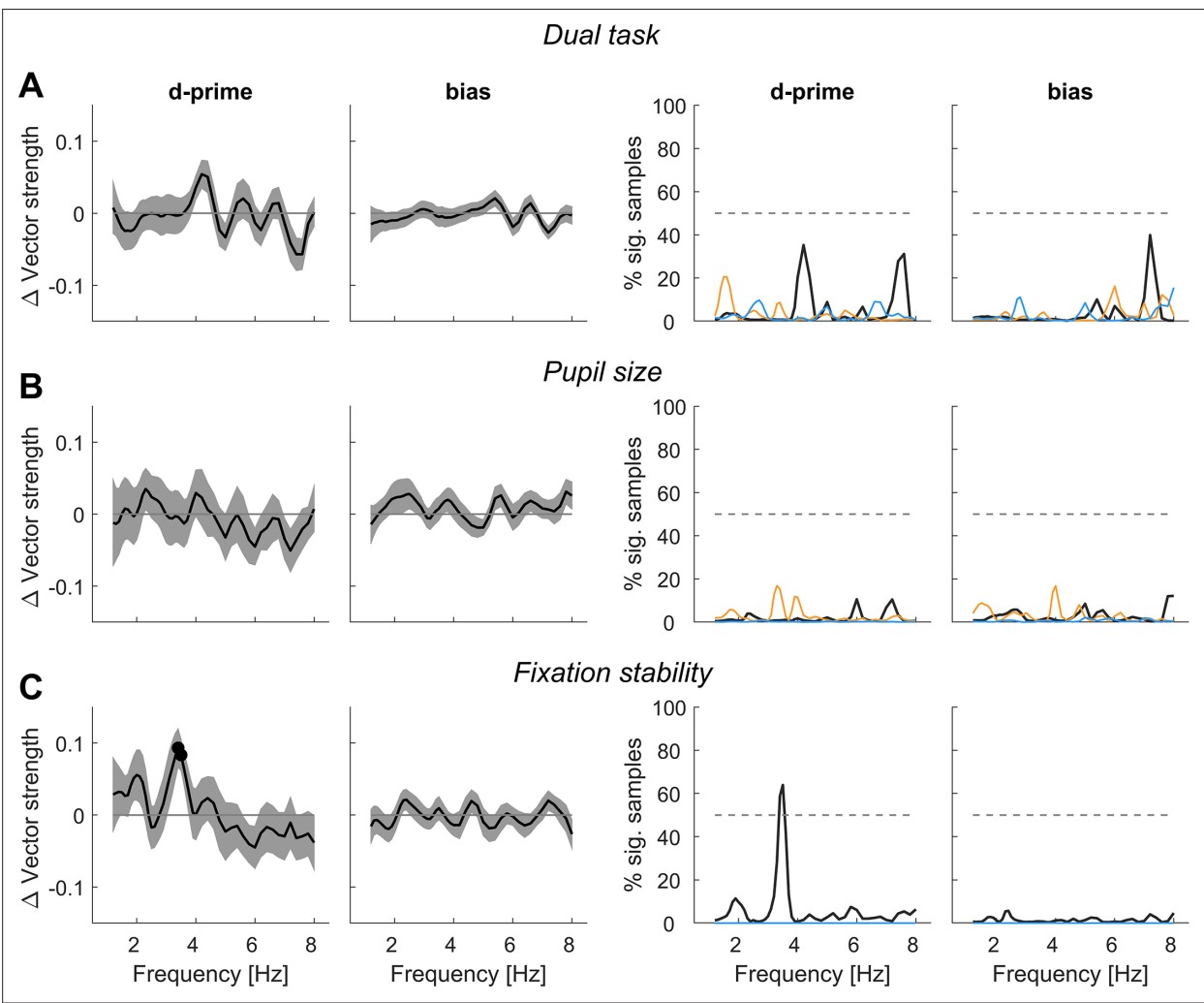

**Figure 8.** Influence of dual task and oculomotor behaviour on rhythmicity in experiment 4. Evidence for rhythmicity was obtained as the group-average vector strength in the model-based analysis of delay-binned data. This evidence was compared between conditions with the graphs showing the difference (mean, s.e.m.). (**A**) Comparison of blocks with dual task those without (n=36). (**B**) Comparison of trials with large pupil diameter minus trials with small diameter (n=24). (**C**) Comparison of trials with less fixation stability (more eye mobility) minus trials with more stability (less eye mobility, n=24). The left panels show the group-level difference (mean, s.e.m.) across the collected participant sample for the combined-ear data. Dots indicate significant effects (paired t-tests, corrected for multiple tests across frequencies at $p<0.05$; for d-prime in panel C, these are 3.4–3.5 Hz). The right panels show the prevalence of effects across random variations in the participant sample for both the combined-ear and individual data.

that judgement (bias 0.08±0.04). Reaction times (with dual task 1.16+0.01 ms vs. without 1.04±0.01, t=11.3, $p<10^{-10}$) and participants' bias towards one judgement differed significantly between blocks with and without the dual task (bias: 0.07±0.05 to -0.03±0.04, t=3.0, p=0.008), while sensitivity to the tone frequencies did not (d-prime: 1.52±0.06 vs 1.47±0.06, t=1.2, p=0.21). To test for an effect of dual task on the strength of rhythm city in behaviour, we contrasted these between blocks with and without dual task. *Figure 8A* shows the result for the combined ear data: the left panel shows the difference in vector strength of rhythmic effects between blocks with and without dual task for the specific participant sample, the right panels show the prevalence of significant effects for individual ears and the combined-ear data derived using resampling. The direct statistical comparison between conditions revealed no significant difference (at *p*<0.05 for the combined-ear data; paired t-tests, corrected for multiple tests across frequencies), and the prevalence of significant effects in variations of the participant sample was low for the unilateral data. However, for the sensitivity in the combined ear data the prevalence of significant effects was nearly 40% around 4 Hz and 8 Hz, suggesting that at these frequencies a subgroup of participants tends to exhibit more rhythmicity when performing the dual task.

When splitting trials by trial-wise pupil dilation, we found no significant difference between trials with larger or smaller pupil size prior to the target in the specific participant sample (*Figure 8B*; left panels). In support of no relation between pupil dilation and rhythmicity, we also found that the prevalence of significant effects was very low (right panels). However, when splitting the data by the trial-wise fixation stability, we found that trials with less fixation stability had significantly stronger rhythmic effect sizes for sensitivity around 3.5 Hz than trials with more fixation stability (at *p*<0.05; *Figure 8C*). Furthermore, the prevalence of significant effects in the combined-ear data among participant samples was higher (64% at 3.5 Hz). This suggests that rhythmicity in behaviour seems to be more expressed when overall eye mobility is larger and underscores an influence of (oculo-)motor behaviour. Here, fixation stability was characterised by the overall eye position, and included both potential saccadic eye movements as well as drifts and microsaccades. The number of actual saccadic eye movements was low and amounted to an average of 0.79±0.1 (mean ± s.e.m) per trial. Hence, any influence of eye mobility is likely linked to microsaccadic behaviour or drifts and not overt large saccades.

## Effects of adaptive task difficulty

The experiments relied on a design in which the task difficulty was maintained around a similar level throughout the experiment. This was implemented by adjusting, where necessary, the stimulus parameters to maintain a comparable level of correct responses (between 69 and 80% on average over 30 trials). We implemented this adaptive task difficulty, as otherwise the behavioural metrics to be analysed (e.g. sensitivity) may be systematically influenced by learning or fatigue, and in the worst case individual participant's performance may either reach chance of near-ceiling levels. This would be counterproductive for the present purpose, as low attentional demands may reduce any influence of rhythmicity in processes such as attention. While the individual participant and ear-specific time courses of stimulus parameters exhibited some fluctuations and sometimes trends reflecting learning, at the group-level these did not differ systematically between the start and end of each experiment for experiment 1 (mean ± s.e.m. change in parameters −1.13±1.46 dB, t=−0.77, p=0.45), experiment 2 (-0.88±2.17 dB, t=−0.40, p=0.68) and experiment 4 (0.08±0.04, t=1.90, p=0.56), while the stimulus intensity was significantly reduced.

## Discussion

The presence of rhythmicity in auditory perceptual judgements remains controversial. We here probed for such rhythmicity in experiments devoid of explicitly entraining sounds. Across data from four experiments and using different analysis approaches, we provide some, but limited, evidence for rhythmicity in listening behaviour. These effects tend to prevail at different frequencies for each ear, and the precise nature of effects differs between experiments, hence suggesting a possible ear- and paradigm-specificity of the putative rhythmicity of auditory perception. At the same time, the prevalence of significant effects in random samples of participants were mostly below 50%, raising questions as to the ubiquity of such effects.

## Assumptions about the specific origin of rhythmicity

One central but often implicit assumption concerns the origin of rhythmicity along the neural pathways. Prominent in vivo recordings have revealed rhythmic activity in the auditory cortex, and hence presumably in areas featuring representations of inputs from both ears (*Lakatos et al., 2016*; *Lakatos et al., 2005*; *Lakatos et al., 2013*; *Kayser et al., 2015*; *Guo et al., 2017*). However, rhythmic activity may also be present in the auditory thalamus, and can even be seen in cochlear recordings (*Gehmacher et al., 2022*; *Köhler and Weisz, 2023*). At the same time, rhythmic activity is also prominent in amodal brain regions involved in decision making, such as the prefrontal cortex (*Samaha et al., 2020*; *Maris et al., 2016*). Any rhythmicity seen in behavioural data could hence originate from neural processes at a single processing stage, or from multiple stages, and could originate from neural processes sensitive only to monaural signals or after binaural integration (*Figure 1A*). A difference between ears could be expected for multiple reasons: first, there are well-known hemisphere differences in the prominence of rhythmic activity in auditory cortices (*Gross et al., 2013*; *Keitel et al., 2018*; *Albouy et al., 2020*; *Flinker et al., 2019*), which could translate into differences in rhythmic perceptual sensitivity for each ear. Second, there is the so-called right-ear advantage, which presumably results from structural asymmetries along auditory pathways (*Kimura, 1967*; *Tanaka et al., 2021*) and which again is presumably associated with hemisphere differences in rhythmic neural activity (*Payne et al., 2017*).

Importantly, different origins have different implications for how rhythmic effects should manifest in the data. Separate origins in monaural neural representations could, in principle, result in effects at different frequencies for each ear (*Figure 1A*, left panel). If this was the case, one may predict that if the data were combined across both ears (i.e. collapsing trials with targets in left and right ears), the evidence for rhythmicity may diminish because incommensurable frequencies would cancel. Furthermore, one may not find rhythmicity when using binaural targets, as they would tap into mechanisms operating at distinct time scales that may cancel during subsequent processing stages. Alternatively, if rhythmicity originates at a single stage after binaural integration, one could expect effects at the same frequency when testing individual ears, when presenting stimuli binaurally or when combining the data across ears (*Figure 1A*, right panel). Hence, depending on the origin of rhythmicity, the choice of monaural or binaural stimuli may dictate which or whether effects can actually be observed.

Assuming rhythmic effects exist in perceptual data, a further point concerns the consistency of their phase within trials of one participant, or between participants in general. The peripheral auditory system is temporally precise, and if the relevant rhythmic processes were driven by acoustic cues such as the onset of the background sound, one should expect these to be tightly aligned in time across trials, at least when considered on time scales such as delta, theta, or alpha. How, if the relevant neurophysiological processes related to more high-level processes such as attention, which fluctuate on slower time scales, it may well be that they are differentially aligned between participants, and even across trials within a participant given changes in vigilance or arousal over the course of the experiment. Importantly, when calculating the time-binned average performance within a participant for analysis, we make the critical assumption of a precise alignment across trials. To what degree this is warranted cannot be clearly concluded at present given the various potential origins of perceptual rhythmicity.

Each of the analysed metrics (sensitivity, bias, and reaction time) reflects a different facet of behaviour. Perceptual sensitivity reflects how well external sensory information is encoded along the sensory and decision pathways, and how reliably this neural representation is translated into motor behaviour. Rhythmic changes in sensitivity may not necessarily imply that the underlying neurophysiological processes are genuinely concerned with the transformation of the sensory signals. For example, rhythmicity in distracting processes that are not primarily concerned with the encoding of the task-relevant information could rhythmically distort the sensory or decision processes and result in measured differences in how precise a stimulus was judged. Rhythmic fluctuations in the perceptual bias, in contrast, could either result from a motor bias or a systematic bias in how sensory signals are directly encoded. The former may relate to rhythmic fluctuations in motor control, while the latter could directly arise from neurophysiological processes carrying the task-relevant signal, which are biased in a relevant representation towards either option. Reaction times, finally, would be rhythmically altered based on changes in neural excitation or signal propagation at essentially any stage between sensory transduction and muscle activity.

## Rhythmic effects emerge at multiple frequencies and independently in each ear

With this in mind, we probed monaural target sounds and performed the analysis both on the data from individual ears and for both ears combined. Overall, we find evidence that rhythmicity in auditory tasks exists. Importantly, the data suggest that rhythmic effects may indeed emerge at distinct frequencies for each ear. In experiments 1 and 2, the evidence for individual ears was not accompanied by concomitant evidence in the combined-ear data, hence suggesting that these effects arise from ear-specific processes operating at different frequencies. Such representations may either relate to monaural auditory representations or representations strongly modulated by spatial attention, and their precise origin needs to be investigated in future work. The ear-specificity of the underlying processes may in part explain the diverging conclusions drawn from studies relying on binaural targets and suggests that rhythmic modes of hearing are better investigated using monaural sounds. A central question for future work will also be to better understand whether the effects reflected at a specific frequency and ear are related to each other, or whether they reflect independent phenomena. It is possible that specific spectral peaks emerge only in a sub-sample of participants and our analysis directly suggests that for a given experiment, the effects in the left and right ears are not correlated among samples of participants. Hence, while rhythmicity seems to exist at different frequencies in the left and right ears, it is possible that these effects prevail, at least partly, in distinct parts of the population.

The present data support conclusions from a previous study using the same experimental paradigm. That study reported effects at different frequencies for perceptual sensitivity and bias, and at slightly different frequencies for each ear (*Ho et al., 2017*). However, in this previous study, the frequencies for both ears fell in a comparable frequency range (6–8 Hz), while in the present data, they differ to a larger degree (varying from 2 to about 6 Hz). One reason for this discrepancy may be that the study by Ho et al. only reported effects for frequencies above 4 Hz and hence may have missed rhythmicity at slower time scales based on the narrower frequency range of interest.

Neural processes at the just-mentioned different time scales have been associated with distinct functions in hearing. Auditory delta band activity has been implied in acoustic filtering of attended information and the task-relevant engagement of auditory networks (*Lakatos et al., 2016*; *Lakatos et al., 2013*; *O'Connell et al., 2014*) and was speculated to reflect one prominent rhythmic mode of listening (*Zoefel and VanRullen, 2017*; *Haegens and Zion Golumbic, 2018*). In contrast, alpha band activity has been linked to task engagement and spatial attention (*Obleser and Kayser, 2019*; *Ng et al., 2012*; *Henry and Obleser, 2012*; *Henry and Herrmann, 2014*). Given that the prevalence of rhythmicity across experiments exhibits multiple peaks, one may conclude that multiple processes shape the rhythmicity of hearing, with their respective prevalence depending on the precise task and participant.

In experiment 3, we found an effect in sensitivity when combining data across ears and a weaker prevalence for an effect in the left ear only. This could speak in favour of an origin in binaural regions, with the single-ear data perhaps not being sufficiently powerful to reach statistical significance. Using the same paradigm as in experiment 3, we have previously shown that the power of EEG-derived oscillatory activity at a time scale of 2–4 Hz modulates the strength by which auditory regions encode the respective evidence about the stimulus sequence (*Kayser et al., 2016*). Such rhythmicity in neural processes supposedly arising from auditory regions may directly relate to the rhythmicity in the perceptual sensitivity observed here. The difference in results between experiments 1 and 3 further underscores the influence of task demands on the apparent rhythmicity in behaviour.

## Role of attention and oculomotor activity

The notion of a rhythmic listening mode has been introduced under the framework of active sensing, whereby perception engages motor routines like eye movements or sniffing to collect sensory information (*Lima et al., 2016*; *De Kock et al., 2021*; *Schroeder et al., 2010*; *Hatsopoulos and Suminski, 2011*). Behavioural studies have shown that making active movements can facilitate listening outcomes, and suggest that the sampling capacity of the auditory system may derive from the motor system (*Morillon et al., 2015*; *Lima et al., 2016*; *De Kock et al., 2021*; *Morillon et al., 2014*; *Zalta et al., 2020*). In fact, corollary signals about oculomotor behaviour are introduced early along the auditory pathway (*Gruters et al., 2018*; *Schmehl and Groh, 2021*), modulate neural responses in the

auditory midbrain (*Bulkin and Groh, 2012*; *Populin et al., 2004*) and cortex (*O'Connell et al., 2020*; *Werner-Reiss et al., 2003*; *Leszczynski et al., 2023*) and patterns of eye movements change during effortful listening (*Contadini-Wright et al., 2023*; *Herrmann and Ryan, 2024*).

Motivated by this, we tested whether the prominence of a rhythmic mode differs when the same task is performed based on automatically paced trials or requires the manual initialisation of each stimulus by the participant (experiment 1 vs 2). The diverging prevalence of effects in these experiments, which differ in time scale and relevant metrics, corroborates an influence of stimulus initialisation on rhythmicity. We did not collect data on the handedness of the participants and future studies would need to replicate this finding and determine whether the lateralisation of effects bears any relation to the manual action when initializing the trial. Using eye tracking data, we also directly asked whether oculomotor activity prior to the target relates to the strength of rhythmicity. We found that more oculomotor activity is associated with greater rhythmicity of perceptual sensitivity around 3–4 Hz. Given that previous studies have found reduced microsaccadic activity during auditory tasks with higher (compared to lower) task demands (*Contadini-Wright et al., 2023*; *Herrmann and Ryan, 2024*), one could interpret this as reflecting influences of momentary task demands on rhythmic processes. While such interpretations remain speculative, the results still further support the notion that hearing is an active process tied to other active and motor behaviour (*Schneider and Mooney, 2015*).

Previous studies have speculated that the auditory system may either operate in a continuous or a rhythmic mode, depending on the current requirements for the task (*Schroeder and Lakatos, 2009*; *Lakatos et al., 2016*; *Zoefel and VanRullen, 2017*). The former is supposedly engaged when high levels of attention are paid to hearing, such as under sustained vigilance. The latter, in contrast, becomes engaged when attention is divided across multiple stimuli or modalities, or when attention is primed in time by virtue of a cue resetting the phase of ongoing rhythmic processes (*Obleser and Kayser, 2019*; *Schroeder and Lakatos, 2009*; *Lakatos et al., 2016*). In the present experiments, there was no sound with explicit and ongoing entraining temporal structure. Rather, attention to the stimulus was cued only by the onset of the background noise, and to a weaker degree, by the preceding fixation period. The latter, however, had a variable duration, and hence presumably a weak influence, if at all. Still, under such conditions, previous studies have reported an apparent rhythmicity of auditory perception (*Ho et al., 2019*; *Ho et al., 2017*; *Hickok et al., 2015*; *Farahbod et al., 2020*; *Zoefel and VanRullen, 2017*; *Ng et al., 2012*), possibly arising from a resetting or restructuring of ongoing rhythmic processes at the delta, theta or alpha band time scale. Experiments probing for rhythmicity in auditory perception in the absence of any sound have failed to show conclusive results (*Zoefel and Heil, 2013*), suggesting that those rhythmic processes shaping auditory perception may be only engaged when auditory pathways are activated by some acoustic input (*Zoefel and VanRullen, 2017*). Possibly, an experimental design with longer trials, that require prolonged continuous vigilance towards the auditory environment, contrasting this divided attention, would yield clearer insights into whether a duality of listening modes can be clearly characterised based on behavioural data.

In experiment 4, we aimed to specifically engage this duality of listening modes using a dual task requiring participants to pay attention to acoustic targets and the fixation dot. While we did not observe significant differences between blocks for the specific participant sample, the prevalence data point to a difference around 4 Hz, with dual-task requirements leading to stronger rhythmicity in perceptual sensitivity. Hence, the present data to lend some support that attention-related task demands do shape a rhythmic listening mode, as speculated previously (*Zoefel and VanRullen, 2017*; *Zoefel and Heil, 2013*; *Lui et al., 2024*).

## Analytical approaches to study rhythmicity

Previous studies have used different analytical approaches to test for rhythmicity and have debated the advantages and disadvantages of these. The individual approaches differ in their implicit assumptions, such as how a null distribution under the assumption of no rhythmic effects is derived (*Ho et al., 2017*; *Kayser, 2019*; *Zoefel et al., 2019*; *Tosato et al., 2022*; *Brookshire, 2022*). They may also differ in their statistical sensitivity and specificity, although for specific studies, these remain often unknown. We implemented multiple approaches, relying on different ways to calculate a measure of effect size for rhythmicity and the respective null distribution. We calibrated these on simulated data, with the aim of reporting effects that are robust to the specific choice of approach. However, the results on the actual data exhibit considerable heterogeneity. And while for the simulated data, the

single-trial approach was most sensitive, this had the lowest prevalence of effects for the actual data. One potential explanation is that the nature of the simulated data differs from that of the actual data, making the former a suboptimal benchmark for the latter. However, given the absence of reliable and reproducible data on the rhythmicity in behavioural data, establishing a proper benchmark remains difficult. Given the (dis-)advantages of specific methods, the present results do not lend themselves for clear conclusions on the suitability of the three approaches.

Importantly, results obtained from a single approach may be misleading, in particular if the inference is drawn from a single statistical threshold applied to one specific participant sample. Testing for the prevalence of effects in random variations of the participant sample did, in part, alleviate some of the discrepancies between approaches in the present data. In fact, given the emergence of multiple spectral peaks in the prevalence data, the results suggest that inference drawn from a single participant sample fails to provide the full diversity of effects present in a population. Hence, it is conceivable that some of the discrepancies in the previous literature are related to statistical noise resulting from small sample sizes and the peculiarities of specific participant samples or analysis approaches.

## Conclusion

In our view, the present data speak against the presence of a mechanism that results in mandatory rhythmicity that governs auditory perception per se. Rather, the present data support the existence of paradigm-specific effects that pertain to sensitivity or response biases independently and which may emerge at different frequencies for each ear. Still, also the present study cannot ultimately prove or refute the existence of a genuine rhythmicity in hearing, demonstrating the methodological difficulties in addressing this question. In line with other recent work, we speculate that multiple processes, including temporal entrainment, neural adaptation, temporally predictive processes, and motor-related processes interact to shape auditory perception (*L'Hermite and Zoefel, 2023*). Given that specific paradigms tap into a unique combination of perceptual and motor requirements, this may explain the diversity of previous results on the rhythmicity of hearing.

# Materials and methods

**Key resources table**

| Reagent type (species) or resource | Designation | Source or reference | Identifiers | Additional information |
|---|---|---|---|---|
| Software, algorithm | Matlab | Mathworks | | Version R2022a |

## Participants and sample size

We collected data from four experiments in which adult volunteers participated after providing informed consent. All participants had self-reported normal vision and hearing and none indicated a history of neurological disorders. Data collection was anonymous and it is possible that some individuals participated in more than one of the experiments. Participants were compensated for their time and the procedures were approved by the ethics committee of Bielefeld University (Nr. 2024–037). We set the a priori sample size for each experiment to n=25. Due to parallel data collection, the actual sample sizes for experiments 1–3 were slightly higher. For experiment 4, we collected more data, in part due to technical problems with also collecting eye tracking data (see below).

## General procedures

The experiments were performed in a darkened and soundproof booth (E: Box; Desone, Germany). Participants sat in front of a computer monitor (27″ monitor; ASUS PG279Q, about 1 m from participant's head) on which visual stimuli were presented. Acoustic stimuli were presented over headphones (Sennheiser DH200Pro). Stimulus presentation was controlled using the Psychophysics Toolbox (Version 3.0.14) using MATLAB (Version R2017a; The MathWorks, Inc, Natick, MA). Participants responded using a computer keyboard. The loudness of the acoustic stimuli was calibrated using a sound level metre (Model 2250 Bruel & Kjær, Denmark).

## Experimental paradigms

The experimental paradigms were modelled based on previous studies and involved the discrimination of monaurally presented target sounds presented among binaural white noise backgrounds

(*Figure 1B*). Targets were presented at different delays relative to the onset of the background in order to probe the influence of this delay on behavioural performance. The individual experiments differed in that the sensory information necessary to perform the task had either to be maintained between trials (discrimination of tones as 'high' or 'low'; experiments 1, 2, 4) or that the discrimination was based on the relative difference of two stimuli presented within a given trial (experiment 3). We also varied whether stimuli were presented at a pseudo-automatic pace or whether stimulus presentation was initialised by the participant (experiment 1 vs experiment 2), and we compared the same experiment when participants focused solely on the auditory task or performed a dual task (experiment 1 vs experiment 4). Participants were instructed to respond as 'fast and accurately' as possible.

Experiment 1 was based on the study by *Ho et al., 2017* and required the categorisation of a target tone as either 'high' or 'low'. Target tones lasted 40 ms (with 5 ms cosine ramps) and had frequencies of 2048 Hz and 1024 Hz. Tones were presented in one ear, at an intensity that was adjusted for each participant and ear to achieve a comparable performance (percent correct responses) between ears and tone frequencies. Target tones were presented at random delays between 300 ms and 1500 ms from the onset of the background noise (sampled uniformly). Background noises were generated on each trial independently for each ear and had an r.m.s level of 65 dB SPL. The target intensity was determined for each participant prior to the experiment using one 3-down 1-up staircase per ear and frequency (with these four staircases presented in an interleaved manner). Thresholds were obtained from the last five reversals. During the actual experiment, tone intensities were updated to maintain performance between 69% and 80% correct responses (determined over the preceding 30 trials) by adjusting the threshold by an amount of 4% of the current value. We implemented this adaptive task difficulty, as otherwise the behavioural metrics to be analysed (e.g. sensitivity) may be systematically influenced by learning or fatigue, which in turn may possibly confound rhythmicity in behaviour. In particular, performance close to chance or ceiling levels may not leave sufficient variability to see any rhythmicity. Each participant performed 8 blocks of 190 trials in one session, resulting in 380 trials per frequency and ear. Each of these 380 trials presented the target at a different (uniformly sampled) delay from background onset. Inter-trial intervals lasted between 1100 ms and 1900 ms (uniform) and the start of each trial was indicated by the appearance of a central fixation dot, with the background noise starting 300 to 500 ms (uniform) after the appearance of the dot. The presentation of the background noise was stopped once participants responded, or latest maximally 1800 ms. For this experiment, we collected data from 27 participants. The response keys (left and right arrow keys) for high and low responses were counterbalanced across participants.

Experiment 2 was similar to experiment 1, except that participants initialised the sound presentation for each trial by pressing a button on the keyboard. Trials started with the presentation of a fixation dot, subsequently to which participants had to press any button to continue the trial. The presentation of the background noise started immediately after registering this button press. For this experiment, we collected data from 26 participants.

Experiment 3 consisted of a within-trial pitch discrimination task modelled based on previous work (*Kayser et al., 2016*). Background noises were as in experiment 1 but target stimuli consisted of two 30 ms tones separated by 40 ms and presented at 60 dB SPL. Participants' task was to determine which tone had higher pitch ('first' or 'second'). Task difficulty was set for each ear and participant by adjusting the frequency difference of the tones (keeping the reference fixed at 1024 Hz). This frequency difference was determined prior to the main experiment using two 3-down 1-up staircases for each ear; it was maintained during the experiment by updating this difference to maintain performance between 69% and 80% correct responses (determined over the last 30 trials) by adjusting the difference by an amount of 4% of the current value. Inter-trial intervals, fixation periods, and the sampling of delays between background onset and target tones (here defined as delay to the start of the first target tone) were as in experiment 1. Other than in experiment 1, response keys were fixed for each participant (left arrow corresponding to first tone being higher, right arrow to second tones). Participants performed eight blocks of 190 trials and we collected data from 26 participants.

Experiment 4 was based on experiment 1 but was designed to probe the role of divided attention. For half the blocks, the design was identical to experiment 1, but the other half required participants to perform the auditory task and a dual task on a central fixation dot. During half the trials, the intensity of this dot changed at a random time and participants had to report whether they perceived this change. This judgement was made subsequent to the auditory judgement. The intensity change was

implemented by either increasing or decreasing the RGB values at a random interval between 100 ms from to background onset to the target (initial values [200, 80, 80], with an in- or decrement of 50). The responses to this dual task were made using an orthogonal axis to that used for the auditory task (using the up and down arrows for 'yes' and 'no' responses, counterbalanced across participants). Each participant performed 16 blocks comprising 130 trials, split over two sessions taking place on different days. In each session, four blocks involved the dual task, with counterbalanced order across the two sessions. This resulted in 260 trials for each tone frequency, ear, and task design (dual task, no dual task), with each trial probing a different delay between background onset and target. In addition to probing the role of the dual task, we also recorded eye movements in experiment 4. Because of technical difficulties in obtaining stable eye movement recordings in some participants and sessions (e.g. resulting from reflections on lenses, etc.), we collected data from a total of 36 participants.

Eye tracking data were recorded from the left eye using an EyeLink 1000 plus eye-tracking system (SR Research) with sampling rate of 500 Hz (for 18 participants) or 2000 Hz (17 participants). Eye-tracking calibration was performed at the beginning of each block using a 9-point grid. The parameters for saccade detection in the EyeLink system ('cognitive' setting) were a velocity threshold of 30°/s and an acceleration threshold of 8000°/s. Given that we here used the eye tracking to characterise fixation stability and pupil size, we combined data obtained with both sampling rates.

We investigated the change of the task difficulty over the course of the experiments. For this, we inspected the time course of the stimulus parameters (e.g. intensity for each tone frequency and ear in experiment 1) for individual participants. For statistical analysis, we converted the stimulus parameter, which is proportional to the voltage associated with each sound, into decibels and computed for each participant the difference between the initial and final value, averaged over ears and frequencies.

## Data preparation

Outliers in the behavioural data were determined as trials with very short (<150 ms) or long reaction times (>2.5 s). We also removed (very rare) trials on which participants pressed a button not assigned to the task. Effectively, we retained 1424±17 (mean ± s.e.m.) for experiment 1, 1435±11 for experiment 2, 1490±8 for experiment 3, and 1865±23 for experiment 4. For subsequent analysis, we transformed reaction times using the square-root transform.

For the analysis of the eye tracking data, we proceeded as follows. We determined blinks and periods containing noisy data (usually arising when participants' eye was directed outside a ± 14° window on either horizontal or vertical axis). We then epoched the data in an interval of –0.4 s to +1.5 s around background onset and retained only trials not involving any of these artefacts at any time point in this epoch. We then retained only those participants with at least 600 trials with good behavioural and eye-tracking data. This was the case for 30 participants (with 1216±67 trials on average). To link the eye data to the rhythmicity in behaviour (see below), we split the trials by either the pupil size or the stability of fixation during each trial. Pupil data were z-scored within each participant over all available epochs. We then calculated for each trial the pupil size as the average size in the interval between the onset of the background noise to the target. Fixation stability was determined as the arithmetic mean of the standard deviations of the eye position along the horizontal and vertical axes in this time interval.

## Overall analysis strategy

The main focus of this study was to probe for signatures of rhythmicity in the behavioural data as a function of the delay between the time of target presentation relative to the onset of the background noise. This was probed using the data of individual participants, once by combining trials regardless of the ear on which the target was presented and separately for trials in each ear. This main question was probed using the data from experiments 1–3, while the data from experiment 4 were used to probe the impact of dual task and the eye properties (see 'Analysis of an influence of dual task and eye metrics').

Given that previous studies differ in how they obtained statistical evidence for rhythmicity in behavioural data and in the metric that was analysed (e.g. response accuracy, reaction time, or sensitivity and bias derived from signal detection theory), we aimed for a comprehensive approach that covered multiple metrics and analyses approaches. In particular, we implemented one analysis line in which the delay was binned into a number of equally spaced bins. Such binning by the variable of

interest allows the calculation of behavioural metrics that are only defined across multiple trials (e.g. sensitivity and bias obtained using signal detection theory). However, binning data by the variable of interest can also distort and lead to spurious effects (*Dowse and Ringo, 1989*; *Dowse and Ringo, 1994*). To avoid such pitfalls and to exploit the full potential of single-trial data, we implemented a separate approach that focused on response accuracy and reaction times of individual trials.

To quantify the strength of putative rhythmicity in the data, we relied on two statistical approaches. In one, we calculated the frequency spectrum of the (delay-binned) data and compared this spectrum to a surrogate distribution. In a second approach, we characterised the data using linear models to separate rhythmic components from non-rhythmic ones.

Given that it is difficult to a priori arbitrate between the different analysis approaches, and given the debate in the literature, we opted for a comprehensive study using each of these. This allowed us to characterise the data using metrics from signal detection theory by combining data within delay bins but also to probe for rhythmicity in the single-trial data. Before explaining the approaches in detail, we summarise the approaches briefly:

1. Based on delay-binned data, we probed rhythmicity in sensitivity, bias, and reaction times using the respective frequency spectra, contrasting the actual data with surrogate data obtained using auto-regressive models (termed Spectral approach in the following).
2. Based on delay-binned data, we probed rhythmicity in sensitivity, bias, and reaction times using linear models separating rhythmic from non-rhythmic predictors. We contrasted the vector strength of the rhythmic predictor with surrogate data obtained using a shuffling procedure (termed Binned approach in the following).
3. Based on the single-trial data, we probed rhythmicity in response accuracy and reaction times using linear models separating rhythmic from non-rhythmic predictors. We contrasted the vector strength of the rhythmic predictor with surrogate data obtained using a shuffling procedure (termed Trial-based approach).

Each approach probes for rhythmicity under the assumption that for a given experiment and analysis, the sample of participants exhibits rhythmicity at the same frequency. However, they do not assume that each participant exhibits rhythmicity at the same phase.

For each approach, we tested 'whether there is rhythmicity at any of the tested frequencies. We opted for this approach as previous studies have reported rhythmicity at frequencies ranging from the delta to the alpha band and hence, no unique and frequency-specific hypothesis can easily be derived from the previous literature. Given that this involves multiple tests across frequencies, and given that each of the approaches may have a different statistical sensitivity and false positive rate, we relied on simulations to calibrate the false positive rate between approaches (see 'Simulations'). This also allowed us to determine their sensitivity on simulated data. To draw any inference from these analyses, we investigated both the probability of observing significant effects in the specific participant samples recruited, corresponding to a classical frequentist approach testing for effects on a fixed sample of participants (*Figure 4*). However, each collected participant sample is only one of many potential samples that could be derived from the entire population (*Kulesa et al., 2015*). Hence, we also explored the within-sample variability of the associated effects using bootstrapping to derive the prevalence of significant effects across variations in the participants (*Figure 5*; see 'Bootstrapping to determine the within-sample variability').

## Spectral analysis of time-binned data

As a first step, we binned the data by the effective delay between background onset and target. We relied on equally spaced bins of 60 ms duration, resulting in 20 bins covering delays from 0 to 1200 ms. Based on the trials in each bin, we computed sensitivity and bias using signal detection theory and the average (square-root transformed) reaction times. We then computed the spectra of each metric after linearly detrending the data and zero-padding by 30 points on either side. Based on these parameters, frequency spectra were computed at effective frequencies between 1.05 Hz and 8.1 Hz at steps of about 0.2 Hz (resulting in similar frequencies as used for the other approaches below).

We then compared the group-averaged spectra to a distribution of surrogate spectra following suggestions in the literature (*Brookshire, 2022*; *Harris and Beale, 2024*). These were obtained for each participant and metric by generating a distribution of 10,000 spectra based on auto-regressive processes of order 1. The parameters estimate for the respective model and simulations

were implemented using the ARfit toolbox in Matlab (*Schneider, 2006*). During AR-model prediction, the first 2000 samples were each ignored. The surrogate spectra were averaged over participants, resulting in a distribution of surrogate data based on which the p-value of the actual group spectrum was computed for each frequency. These p-values were derived using a percentile bootstrap approach, by comparing the actual group average to the distribution of group-averages in the surrogate data. These first-level p-values were then thresholded at an appropriate level to achieve a desired false positive rate across analysis approaches (see 'Simulations'). Note that there are slight variations of how AR surrogate spectra can be used to test for rhythmicity at the group level. We here computed the actual and surrogate spectra separately for each participant as suggested recently (*Harris and Beale, 2024*), but relied on a percentile bootstrap test to compare the group-level actual and surrogate data rather than relying on a parametric comparison, such as by comparing actual and surrogate data using t-tests.

## Linear model-based analysis of time-binned data

We computed sensitivity, bias, and reaction time for the delay-binned data as described above. We then modelled the data using linear models comprising both rhythmic and non-rhythmic predictors as implemented previously (*Kayser, 2019*). Effectively, we described the data using the following terms in a linear model assuming normally distributed data and an identical link function: an offset, a linear influence of delay, a u/v shaped influence reflecting changes in behavioural data tied to the duration of the overall period of target uncertainty (here using a frequency of 0.5 Hz), and a rhythmic predictor consisting of the sine and cosine component at a specific frequency. For each participant, we then derived the vector strength of the rhythmic predictor, defined as the arithmetic mean of the squared betas for the sine and cosine components. This vector strength reflects the overall prominence of the rhythmic predictor.

For statistical testing, we contrasted the group-average vector strength to a surrogate distribution obtained under the null hypothesis of no relation between behavioural data and delay. This was implemented for each participant by shuffling this assignment 10,000 times and calculating the respective model. The resulting vector strengths were averaged across participants, resulting in a distribution of surrogate values. Based on this distribution, we computed the p-value of the actual vector strength based on the percentiles of the surrogate data, again favouring a non-parametric over a parametric approach. Separate models were fit for rhythmic predictors at frequencies between 1.2 Hz and 8 Hz, with steps of 0.1 Hz between 1.2 and 4 Hz and steps of 0.2 Hz above. Again, these first-level p-values were thresholded at an appropriate level to achieve a desired false positive rate.

We note that the spectral- and the model-based approach are related but also distinct. Computing the frequency spectrum effectively describes the data as a superposition of distinct but simultaneously present rhythmic components, while the model-based approach tests one rhythmic predictor at a time. The spectral approach provides a more accurate depiction of the data if rhythmicity exists at several frequencies, as each predictor is considered simultaneously. However, due to spectral blurring, the statistical power at individual frequencies may also be diluted.

## Linear model based on single-trial data

We computed similar linear models as described above, but applied these to the single-trial accuracy and the reaction times as metrics. For reaction times, we relied on a linear model assuming normally distributed data and a linear link function; for accuracy, we relied on a binomial model and a logistic link function. As for the binned data, we compared the actual group-averaged vector strength at individual frequencies to surrogate data. As the single trial data effectively allow a higher temporal resolution compared to the binned data, we here tested frequencies between 1.2 Hz and 12 Hz (with steps of 0.8 Hz between 8 and 12 Hz).

## Visualisation of statistical results

To visualise the evidence for a rhythmic effect for the specific participant sample, we show the associated (log-transformed) first level p-values (*Figure 4*). Although this measure of statistical significance does not reflect a measure of the underlying effect size (e.g. spectral power), it allows presenting the results on a scale that can be directly compared between analysis approaches, metrics, frequencies, and analyses focusing on individual ears or the combined data. Each approach has a different

statistical sensitivity and the underlying effect sizes (e.g. spectral power) vary with frequency for both the actual data and null distribution. As a result, the effect size reaching statistical significance varies with frequency, metrics, and analyses. By showing p-values, we overcome this variability and present the data on a scale where the cut-offs for significance are the same across all dimensions (c.f. lines in *Figure 4*). Still, for comparison, we do show the actual rhythmic effects for the time-binned approach in *Figure 5*. That figure shows the combined vector strength of the two rhythmic predictors for the actual data and the surrogate value corresponding to a significance level of $p<0.05$.

## Calibrating analysis approaches on simulated data

Each approach may differ in the effective sensitivity and specificity of detecting rhythmicity (*Zoefel et al., 2019*; *Tosato et al., 2022*; *Brookshire, 2022*). We hence based the interpretation not on the first-level p-values obtained by the individual comparisons of actual vs. surrogate data. Rather, we simulated data with genuine rhythmicity and data without and calculated sensitivity and specificity of each approach at different signal-to-noise ratios (SNRs). We then selected those first-level p-values that produce a false positive rate (specificity) of 0.05 when detecting a rhythmic effect at any frequencies in the simulated data (i.e. correcting for multiple tests along the frequency axis).

Practically, we simulated data for a sample of 25 participants, with 700 trials each. This corresponds to a sample size and trial count similar to the actual data, matching the analysis of individual ears. We simulated normally distributed data that was generated using a linear model (similar to that used during data analysis): we generated data using a superposition of an offset, a linear slope, a u/v shaped term, and both sine and cosine predictors for the rhythmic component. For each parameter setting (see below), we simulated 1000 samples of participants (i.e. virtual experiments). In each simulation, we drew the single-trial betas for the model generating the data from Gaussian distributions with predefined means and standard deviations independently across participants and simulations. For simulations with a rhythmic effect, the parameters were as follows: offset (1, 0.1; mean, SD of the Gaussian distribution used to draw the single participant values), linear slope (0.1, 0.1), u/V term (0.1, 0.1), sine at 4 Hz (0.2, 0.1), cosine at 4 Hz (0.2, 0.1). For simulations without rhythmic effect, the sine and cosine terms were zeroed. The simulated data were analysed in the same manner as the actual data, resulting in 1000 samples of first-level p-values for each analysis approach.

We implemented simulations using different SNR. These were achieved by adding Gaussian noise to the simulated data, with mean zero but varying SDs (taking values from 2 to 8 for simulations with a rhythmic effect; and values of 2, 4, and 6 for simulations without). We then calculated the fraction of simulations in which a significant effect was detected at any frequency using different first-level p-values as cut-offs. For runs with a simulated rhythmic effect, this yields the statistical sensitivity and for runs without such, this yields the statistical specificity. For each approach, we selected those thresholds for the first-level significance yielding an approximate false positive rate of 0.05 (and separately also for 0.01) across all 3000 runs without effect. These cut-offs are shown together with the actual data in *Figure 4*. *Figure 2* shows the resulting true and false positive rates on simulated data when using a cut-off of $p{\sim}0.05$.

## Resampling to determine the within-sample variability

In addition to presenting the significance of effects for the collected participant sample, we explored the prevalence of effects among random variations in this sample (*Kulesa et al., 2015*). We used bootstrapping to repeatedly draw participant samples from the entire sample (e.g. for experiment 1, we sampled 26 participants at random with replacement from the pool of 26 participants collected). We then determined the percentage of simulations that yield a significant effect at each frequency. These estimates of effect prevalence are shown in *Figure 6* and provide an estimate of the reproducibility of effects across random variations in the participant sample. This approach to investigating the prevalence of a specific (significant) effect in the participant sample has the advantage that it avoids potentially biased conclusions drawn from one single sample of participants and provides a quantification of how consistent an effect is within a population of participants. At the same time, there is no clear single cut-off point describing what prevalence value should be considered as 'important', in contrast to classical p-values. Hence, this approach to quantifying effects is more qualitative rather than binary, similar as when using Bayesian approaches to address scientific hypotheses (*Wagenmakers, 2007*).

## Analysis of an influence of dual task and eye metrics

We used the data of experiment 4 to probe the influence of a dual task on signatures of rhythmicity. For this, we contrasted the data from blocks with the dual task to those without dual task. We also used the data from experiment 4 to probe whether signatures of rhythmicity are related to pupil size as a measure of task engagement (*Joshi and Gold, 2020*; *Zekveld et al., 2018*) or the overall mobility of eyes during the trial (*Joshi and Gold, 2020*; *Zekveld et al., 2018*).

In contrast to the analyses of the data from experiments 1–3, in which we probed the existence of signatures of rhythmicity against a suitable null distribution, we here followed a different logic. The analyses for experiment 4 are based on contrasting two halves of the data, effectively comparing two equivalent signatures for rhythmicity within participants. To derive such a signature of rhythmicity, we relied on the linear model-based analysis of the delay-binned data. For a given set of trials, we derived the group-averaged vector strength of the rhythmic predictor as a function of frequency. These group-level vector strengths were then contrasted between conditions (e.g. trials with or without dual task) using paired t-tests, the p-values of which were corrected for multiple tests across frequencies using the Benjamini & Hochberg procedure (*Figure 7*). As for the analysis of experiments 1–3, we looked at both the significance for the specific available participant sample, and tested for the generalisation of effects among participant samples using bootstrapping. For the latter, we again repeated the analysis using randomly sampled participants, and counted the number of participant samples yielding a significant effect at a specific frequency.

For the data split by dual task, we contrasted the vector strength obtained using the trials from the respective blocks (n=37 participants). For the effect of pupil size and eye mobility, we implemented median splits on the respective variables within each participant. Given that the available number of trials with good eye tracking data varied across participants, and given that this data split further reduces the effective number of trials per condition of interest, we implemented this analysis only for those n=24 participants that had 400 trials per split (average number of trials per split 674±29).

## Acknowledgements

We thank Lena Hehemann, Sepideh Mirzaei, and Stella Thiele for their help with collecting the data. This study was supported by the German Research Foundation (DFG, KA 2661/6–1).

# Additional information

### Funding

| Funder | Grant reference number | Author |
| --- | --- | --- |
| Deutsche Forschungsgemeinschaft | KA 2661/6-1 | Cécile Fabio Christoph Kayser |

The funders had no role in study design, data collection and interpretation, or the decision to submit the work for publication.

### Author contributions

Cécile Fabio, Conceptualization, Data curation, Formal analysis, Investigation, Visualization, Methodology, Writing – original draft, Writing – review and editing; Christoph Kayser, Conceptualization, Software, Formal analysis, Supervision, Funding acquisition, Investigation, Visualization, Methodology, Writing – original draft, Project administration, Writing – review and editing

### Author ORCIDs

Cécile Fabio (ID) https://orcid.org/0000-0003-0573-6378
Christoph Kayser (ID) https://orcid.org/0000-0001-7362-5704

### Ethics

Human subjects: We collected data from four experiments in which adult volunteers participated after providing informed consent. All participants had self-reported normal vision and hearing and none indicated a history of neurological disorders. Data collection was anonymous and it is possible that

some individuals participated in more than one of the experiments. Participants were compensated for their time and the procedures were approved by the ethics committee of Bielefeld University (Nr. 2024-037).

Reviewer #1 (Public review): https://doi.org/10.7554/eLife.105734.3.sa1
Reviewer #2 (Public review): https://doi.org/10.7554/eLife.105734.3.sa2
Reviewer #3 (Public review): https://doi.org/10.7554/eLife.105734.3.sa3
Author response https://doi.org/10.7554/eLife.105734.3.sa4

## Additional files

### Supplementary files
MDAR checklist

### Data availability
Data and code are available on github: https://github.com/christophckayser/Fabio_eLife2025 (copy archived at *Kayser, 2025*).

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
