## [Editor Report · eLife Assessment]

This high-N, multi-task study offers a comprehensive examination of rhythmicity in behavioral performance during listening. It presents a **valuable** set of findings that reveal task- and ear-specific effects, challenging the notion of a universal rhythmicity in auditory perception. The evidence is **solid** and the work is likely to be of significant interest to behavioral and cognitive scientists focused on perception and neural oscillations.

---

## [Referee Report · Reviewer #1 (Public review)]

Summary:

This paper presents results from four independent experiments, each of them testing for rhythmicity in auditory perception. The authors report rhythmic fluctuations in discrimination performance at frequencies between 2 and 6 Hz. The exact frequency depends on the ear and experimental paradigm, although some frequencies seem to be more common than others.

Strengths:

The first sentence in the abstract describes the state of the art perfectly: "Numerous studies advocate for a rhythmic mode of perception; however, the evidence in the context of auditory perception remains inconsistent". This is precisely why the data from the present study is so valuable. This is probably the study with the highest sample size (total of > 100 in 4 experiments) in the field. The analysis is very thorough and transparent, due to the comparison of several statistical approaches and simulations of their sensitivity. Each of the experiments differs from the others in a clearly defined experimental parameter, and the authors test how this impacts auditory rhythmicity, measured in pitch discrimination performance (accuracy, sensitivity, bias) of a target presented at various delays after noise onset.

Weaknesses:

The authors find that the frequency in auditory perception changes between experiments. Possible reasons for such differences are described, but they remain difficult to interpret, as it is unclear whether they merely reflect some natural variability (independent of experimental parameters) or are indeed driven by the specific experimental paradigm (and therefore replicable).

Therefore, it remains to be shown whether there is any systematic pattern in the results that allows conclusions about the roles of different frequencies.

---

## [Referee Report · Reviewer #2 (Public review)]

Summary:

The current study aims to shed light on why previous work on perceptual rhythmicity has led to inconsistent results. They propose that the differences may stem from conceptual and methodological issues. In a series of experiments, the current study reports perceptual rhythmicity in different frequency bands that differ between different ear stimulations and behavioral measures. The study suggests challenges regarding the idea of universal perceptual rhythmicity in hearing.

Strengths:

The study aims to address differences observed in previous studies about perceptual rhythmicity. This is important and timely because the existing literature provides quite inconsistent findings. Several experiments were conducted to assess perceptual rhythmicity in hearing from different angles. The authors use sophisticated approaches to address the research questions. The manuscript has greatly improved after the revision.

Weaknesses:

Additional variance: In several experiments, a fixation cross preceded - at a variable interval - the onset of the background noise that aimed to reset the phase of an ongoing oscillation. There is the chance that the fixation cross also resets the phase, potentially adding variance to the data. In addition, the authors used an adaptive procedure during the experimental blocks such that the stimulus intensity was adjusted throughout. There is good reason for doing so, but it means that correctly identified/discriminated targets will on average have a greater stimulus intensity. This may add variance to the data. These two aspects may potentially contribute to the observation of weak perceptual rhythmicity.

Figures: The text in Figures 4 and 6 is small. I think readers would benefit from a larger font size. Moreover, Figure 1A is not very intuitive. Perhaps it could be made clearer. The new Figure 5 was not discussed in the text. I wonder whether analyses with traditional t-tests could be placed in the supplements.

50% significant samples: The authors consider 50% of significant bootstrapped samples robust. For example: "This revealed that the above‐mentioned effects prevail for at least 50% of the simulated experiments, corroborating their robustness within the participant sample". Many of the effects have even lower than 50% of significant samples. It is a matter of opinion of what is robust or not, but I think combined with the overall variable nature of the effects in different frequency bands and ears etc. leaves more the impression that the effects are not very robust. I think the authors state it correctly in the last sentence of the first paragraph of the discussion: "At the same time the prevalence of significant effects in random samples of participants were mostly below 50%, raising questions as to the ubiquity of such effects." I think the authors should update the abstract in this regard to avoid that readers who only read the abstract get the wrong impression about the robustness of the effects. It is not clear to me if the same study (using the same conditions) was done in a different lab that the results would come out similarly to the results reported here.

---

## [Referee Report · Reviewer #3 (Public review)]

Summary:

The finding of rhythmic activity in the brain has for a long time engendered the theory of rhythmic modes of perception, that humans might oscillate between improved and worse perception depending on states of our internal systems. However, experiments looking for such modes have resulted in conflicting findings, particularly in those where the stimulus itself is not rhythmic. This paper seeks to take a comprehensive look at the effect and various experimental parameters which might generate these competing findings: in particular, the presentation of the stimulus to one ear or the other, the relevance of motor involvement, attentional demands, and memory: each of which are revealed to effect the consistency of this rhythmicity.

The need the paper attempts to resolve is a critical one for the field. However, as presented, I remain unconvinced that the data would not be better interpreted as showing no consistent rhythmic mode effect.

Strengths:

The paper is strong in its experimental protocol and its comprehensive analysis which seeks to compare effects across several analysis types and slight experiment changes to investigate which parameters could effect the presence or absence of an effect of rhythmicity. The prescribed nature of its hypotheses and its manner to set out to test them is very clear which allows for a straightforward assessment of its results

Weaknesses:

The papers cited to justify a rhythmic mode are largely based on the processing of rhythmic stimuli. The authors assume the rhythmic mode to be the general default but its not so clear to me why this would be so. The task design seems better suited to a continuous vigilance mode task.

Secondly, the analysis to detect a "rhythmic mode", assumes a total phase rest at noise onset which is highly implausible given standard nonlinear dynamical analysis of oscillator performance. It's not clear that a rhythmic mode (should it be applied in this task) would indeed generate a consistent phase as the analysis searches for.

Thirdly, the number of statistical tests used here make trusting any single effect quite difficult and very few of the effects replicate more than once. I think the better would be interpreted as not confirming evidence for rhythmic mode processing in the ears.

Comments on revised version:

No further comments. The paper has much of the same issues that I expressed in the initial review but I don't think they can be addressed without a replication study which I appreciate is not always plausible.

---

## [Author Response]

The following is the authors’ response to the original reviews.

**Reviewer #1 (Public review):**
Summary:This paper presents results from four independent experiments, each of which tests for rhythmicity in auditory perception. The authors report rhythmic fluctuations in discrimination performance at frequencies between 2 and 6 Hz. The exact frequency depends on the ear and experimental paradigm, although some frequencies seem to be more common than others.Strengths:The first sentence in the abstract describes the state of the art perfectly: "Numerous studies advocate for a rhythmic mode of perception; however, the evidence in the context of auditory perception remains inconsistent". This is precisely why the data from the present study is so valuable. This is probably the study with the highest sample size (total of > 100 in 4 experiments) in the field. The analysis is very thorough and transparent, due to the comparison of several statistical approaches and simulations of their sensitivity. Each of the experiments differs from the others in a clearly defined experimental parameter, and the authors test how this impacts auditory rhythmicity, measured in pitch discrimination performance (accuracy, sensitivity, bias) of a target presented at various delays after noise onset.Weaknesses:(1) The authors find that the frequency of auditory perception changes between experiments. I think they could exploit differences between experiments better to interpret and understand the obtained results. These differences are very well described in the Introduction, but don't seem to be used for the interpretation of results. For instance, what does it mean if perceptual frequency changes from between- to within-trial pitch discrimination? Why did the authors choose this experimental manipulation? Based on differences between experiments, is there any systematic pattern in the results that allows conclusions about the roles of different frequencies? I think the Discussion would benefit from an extension to cover this aspect.

We believe that interpreting these differences remains difficult and a precise, detailed (and possibly mechanistic) interpretation is beyond the goal of the present study. The main goal of this study was to explore the consistency and variability of effects across variations of the experimental design and samples of participants. Interpreting specific effects, e.g. at particular frequencies, would make sense mostly if differences between experiments have been confirmed in a separate reproduction. Still, we do provide specific arguments for why differences in the outcome between different experiments, e.g. with and without explicit trial initialization by the participants, could be expected. See lines 91ff in the introduction and 786ff in the discussion.

(2) The Results give the impression of clear-cut differences in relevant frequencies between experiments (e.g., 2 Hz in Experiment 1, 6 Hz in Exp 2, etc), but they might not be so different. For instance, a 6 Hz effect is also visible in Experiment 1, but it just does not reach conventional significance. The average across the three experiments is therefore very useful, and also seems to suggest that differences between experiments are not very pronounced (otherwise the average would not produce clear peaks in the spectrum). I suggest making this point clearer in the text.

We have revised the conclusions to note that the present data do not support clear cut differences between experiments. For this reason we also refrain from detailed interpretations of specific effects, as suggested by this reviewer in point 1 above.

(3) I struggle to understand the hypothesis that rhythmic sampling differs between ears. In most everyday scenarios, the same sounds arrive at both ears, and the time difference between the two is too small to play a role for the frequencies tested. If both ears operate at different frequencies, the effects of the rhythm on overall perception would then often cancel out. But if this is the case, why would the two ears have different rhythms to begin with? This could be described in more detail.

This hypothesis was not invented by us, but in essence put forward in previous work. The study by Ho et al. CurrBiol 2017 has reported rhythmic effects at different frequencies in the left and right ears, and we here tried to reproduce these effects. One could speculate about an ear-difference based on studies reporting a right-ear advantage in specific listening tasks, and the idea that different time scales of rhythmic brain activity may be specifically prevail in the left and right cortical hemispheres; hence it does not seem improbable that there could be rhythmic effects in both ears at different frequencies. We note this in the introduction, l. 65ff.

**Reviewer #2 (Public review):**
Summary:The current study aims to shed light on why previous work on perceptual rhythmicity has led to inconsistent results. They propose that the differences may stem from conceptual and methodological issues. In a series of experiments, the current study reports perceptual rhythmicity in different frequency bands that differ between different ear stimulations and behavioral measures.The study suggests challenges regarding the idea of universal perceptual rhythmicity in hearing.Strengths:The study aims to address differences observed in previous studies about perceptual rhythmicity. This is important and timely because the existing literature provides quite inconsistent findings. Several experiments were conducted to assess perceptual rhythmicity in hearing from different angles. The authors use sophisticated approaches to address the research questions.Weaknesses:(1) Conceptional concerns:The authors place their research in the context of a rhythmic mode of perception. They also discuss continuous vs rhythmic mode processing. Their study further follows a design that seems to be based on paradigms that assume a recent phase in neural oscillations that subsequently influence perception (e.g., Fiebelkorn et al.; Landau & Fries). In my view, these are different facets in the neural oscillation research space that require a bit more nuanced separation. Continuous mode processing is associated with vigilance tasks (work by Schroeder and Lakatos; reduction of low frequency oscillations and sustained gamma activity), whereas the authors of this study seem to link it to hearing tasks specifically (e.g., line 694). Rhythmic mode processing is associated with rhythmic stimulation by which neural oscillations entrain and influence perception (also, Schroeder and Lakatos; greater low-frequency fluctuations and more rhythmic gamma activity). The current study mirrors the continuous rather than the rhythmic mode (i.e., there was no rhythmic stimulation), but even the former seems not fully fitting, because trials are 1.8 s short and do not really reflect a vigilance task. Finally, previous paradigms on phase-resetting reflect more closely the design of the current study (i.e., different times of a target stimulus relative to the reset of an oscillation). This is the work by Fiebelkorn et al., Landau & Fries, and others, which do not seem to be cited here, which I find surprising. Moreover, the authors would want to discuss the role of the background noise in resetting the phase of an oscillation, and the role of the fixation cross also possibly resetting the phase of an oscillation. Regardless, the conceptional mixture of all these facets makes interpretations really challenging. The phase-reset nature of the paradigm is not (or not well) explained, and the discussion mixes the different concepts and approaches. I recommend that the authors frame their work more clearly in the context of these different concepts (affecting large portions of the manuscript).

Indeed, the paradigms used here and in many similar previous studies incorporate an aspect of phase-resetting, as the presentation of a background noisy may effectively reset ongoing auditory cortical processes. Studies trying to probe for rhythmicity in auditory perception in the absence any background noise have not shown any effect (Zoefel and Heil, 2013), perhaps because the necessary rhythmic processes along auditory pathways are only engaged when some sound is present. We now discuss these points, and also acknowledge the mentioned studies in the visual system; l. 57.

(2) Methodological concerns:The authors use a relatively unorthodox approach to statistical testing. I understand that they try to capture and characterize the sensitivity of the different analysis approaches to rhythmic behavioral effects. However, it is a bit unclear what meaningful effects are in the study. For example, the bootstrapping approach that identifies the percentage of significant variations of sample selections is rather descriptive (Figures 5-7). The authors seem to suggest that 50% of the samples are meaningful (given the dashed line in the figure), even though this is rarely reached in any of the analyses. Perhaps >80% of samples should show a significant effect to be meaningful (at least to my subjective mind). To me, the low percentage rather suggests that there is not too much meaningful rhythmicity present.

We note that there is no clear consensus on what fraction of experiments should be expected or how this way of quantifying effects should be precisely valued (l. 441ff). However, we now also clearly acknowledge in the discussion that the effective prevalence is not very high (l. 663).

I suggest that the authors also present more traditional, perhaps multi-level, analyses: Calculation of spectra, binning, or single-trial analysis for each participant and condition, and the respective calculation of the surrogate data analysis, and then comparison of the surrogate data to the original data on the second (participant) level using t-tests. I also thought the statistical approach undertaken here could have been a bit more clearly/didactically described as well.

We here realize that our description of the methods was possibly not fully clear. We do follow the strategy as suggested by this reviewer, but rather than comparing actual and surrogate data based on a parametric t-test, we compare these based on a non-parametric percentile-based approach. This has the advantage of not making specific (and possibly not-warranted) assumptions about the distribution of the data. We have revised the methods to clarify this, l. 332ff.

The authors used an adaptive procedure during the experimental blocks such that the stimulus intensity was adjusted throughout. In practice, this can be a disadvantage relative to keeping the intensity constant throughout, because, on average, correct trials will be associated with a higher intensity than incorrect trials, potentially making observations of perceptual rhythmicity more challenging. The authors would want to discuss this potential issue. Intensity adjustments could perhaps contribute to the observed rhythmicity effects. Perhaps the rhythmicity of the stimulus intensity could be analyzed as well. In any case, the adaptive procedure may add variance to the data.

We have added an analysis of task difficulty to the results (new section “Effects of adaptive task difficulty“) to address this. Overall we do not find systematic changes in task difficulty across participants for most of the experiments, but for sure one cannot rule out that this aspect of the design also affects the outcomes. Importantly, we relied on an adaptive task difficulty to actually (or hopefully) reduce variance in the data, by keeping the task-difficulty around a certain level. Give the large number of trials collected, not using such an adaptive produce may result in performance levels around chance or near ceiling, which would make impossible to detect rhythmic variations in behavior.

Additional methodological concerns relate to Figure 8. Figures 8A and C seem to indicate that a baseline correction for a very short time window was calculated (I could not find anything about this in the methods section). The data seem very variable and artificially constrained in the baseline time window. It was unclear what the reader might take from Figure 8.

This figure was intended mostly for illustration of the eye tracking data, but we agree that there is no specific key insight to be taken from this. We removed this.

Motivation and discussion of eye-movement/pupillometry and motor activity: The dual task paradigm of Experiment 4 and the reasons for assessing eye metrics in the current study could have been better motivated. The experiment somehow does not fit in very well. There is recent evidence that eye movements decrease during effortful tasks (e.g., Contadini-Wright et al. 2023 J Neurosci; Herrmann & Ryan 2024 J Cog Neurosci), which appears to contradict the results presented in the current study. Moreover, by appealing to active sensing frameworks, the authors suggest that active movements can facilitate listening outcomes (line 677; they should provide a reference for this claim), but it is unclear how this would relate to eye movements. Certainly, a person may move their head closer to a sound source in the presence of competing sound to increase the signal-to-noise ratio, but this is not really the active movements that are measured here. A more detailed discussion may be important. The authors further frame the difference between Experiments 1 and 2 as being related to participants' motor activity. However, there are other factors that could explain differences between experiments. Self-paced trials give participants the opportunity to rest more (inter-trial durations were likely longer in Experiment 2), perhaps affecting attentional engagement. I think a more nuanced discussion may be warranted.

We expanded the motivation of why self-pacing trials may effectively alter how rhythmic processes affect perception, and now also allude to attention and expectation related effects (l. 786ff). Regarding eye movements we now discuss the results in the light of the previously mentioned studies, but again refrain from a very detailed and mechanistic interpretation (l. 782).

Discussion:The main data in Figure 3 showed little rhythmicity. The authors seem to glance over this fact by simply stating that the same phase is not necessary for their statistical analysis. Previous work, however, showed rhythmicity in the across-participant average (e.g., Fiebelkorn's and similar work). Moreover, one would expect that some of the effects in the low-frequency band (e.g., 2-4 Hz) are somewhat similar across participants. Conduction delays in the auditory system are much smaller than the 0.25-0.5 s associated with 2-4 Hz. The authors would want to discuss why different participants would express so vastly different phases that the across-participant average does not show any rhythmicity, and what this would mean neurophysiologically.

We now discussion the assumptions and implications of similar or distinct phases of rhythmic processes within and between participants (l. 695ff). In particular we note that different origins of the underlying neurophysiological processes eventually may suggest that such assumptions are or a not warranted.

An additional point that may require more nuanced discussion is related to the rhythmicity of response bias versus sensitivity. The authors could discuss what the rhythmicity of these different measures in different frequency bands means, with respect to underlying neural oscillations.

We expanded discussion to interpret what rhythmic changes in each of the behavioral metric could imply (l. 706ff).

Figures:Much of the text in the figures seems really small. Perhaps the authors would want to ensure it is readable even for those with low vision abilities. Moreover, Figure 1A is not as intuitive as it could be and may perhaps be made clearer. I also suggest the authors discuss a bit more the potential monoaural vs binaural issues, because the perceptual rhythmicity is much slower than any conduction delays in the auditory system that could lead to interference.

We tried to improve the font sizes where possible, and discuss the potential monaural origins as suggested by other reviewers.

**Reviewer #3 (Public review):**
Summary:The finding of rhythmic activity in the brain has, for a long time, engendered the theory of rhythmic modes of perception, that humans might oscillate between improved and worse perception depending on states of our internal systems. However, experiments looking for such modes have resulted in conflicting findings, particularly in those where the stimulus itself is not rhythmic. This paper seeks to take a comprehensive look at the effect and various experimental parameters which might generate these competing findings: in particular, the presentation of the stimulus to one ear or the other, the relevance of motor involvement, attentional demands, and memory: each of which are revealed to effect the consistency of this rhythmicity.The need the paper attempts to resolve is a critical one for the field. However, as presented, I remain unconvinced that the data would not be better interpreted as showing no consistent rhythmic mode effect. It lacks a conceptual framework to understand why effects might be consistent in each ear but at different frequencies and only for some tasks with slight variants, some affecting sensitivity and some affecting bias.Strengths:The paper is strong in its experimental protocol and its comprehensive analysis, which seeks to compare effects across several analysis types and slight experiment changes to investigate which parameters could affect the presence or absence of an effect of rhythmicity. The prescribed nature of its hypotheses and its manner of setting out to test them is very clear, which allows for a straightforward assessment of its resultsWeaknesses:There is a weakness throughout the paper in terms of establishing a conceptual framework both for the source of "rhythmic modes" and for the interpretation of the results. Before understanding the data on this matter, it would be useful to discuss why one would posit such a theory to begin with. From a perceptual side, rhythmic modes of processing in the absence of rhythmic stimuli would not appear to provide any benefit to processing. From a biological or homeostatic argument, it's unclear why we would expect such fluctuations to occur in such a narrow-band way when neither the stimulus nor the neurobiological circuits require it.

We believe that the framework for why there may be rhythmic activity along auditory pathways that shapes behavioral outcomes has been laid out in many previous studies, prominently here (Schroeder et al., 2008; Schroeder and Lakatos, 2009; Obleser and Kayser, 2019). Many of the relevant studies are cited in the introduction, which is already rather long given the many points covered in this study.

Secondly, for the analysis to detect a "rhythmic mode", it must assume that the phase of fluctuations across an experiment (i.e., whether fluctuations are in an up-state or down-state at onset) is constant at stimulus onset, whereas most oscillations do not have such a total phase-reset as a result of input. Therefore, some theoretical positing of what kind of mechanism could generate this fluctuation is critical toward understanding whether the analysis is well-suited to the studied mechanism.

In line with this and previous comments (by reviewer 2) we have expanded the discussion to consider the issue of phase alignment (l. 695ff).

Thirdly, an interpretation of why we should expect left and right ears to have distinct frequency ranges of fluctuations is required. There are a large number of statistical tests in this paper, and it's not clear how multiple comparisons are controlled for, apart from experiment 4 (which specifies B&H false discovery rate). As such, one critical method to identify whether the results are not the result of noise or sample-specific biases is the plausibility of the finding. On its face, maintaining distinct frequencies of perception in each ear does not fit an obvious conceptual framework.

Again this point was also noted by another reviewer and we expanded the introduction and discussion in this regard (l. 65ff).

**Reviewer #1 (Recommendations for the authors):**
(1) An update of the AR-surrogate method has recently been published (https://doi.org/10.1101/2024.08.22.609278). I appreciate that this is a lot of work, and it is of coursee up to the authors, but given the higher sensitivity of this method, it might be worth applying it to the four datasets described here.

Reading this article we note that our implementation of the AR-surrogate method was essentially as suggested here, and not as implemented by Brookshire. In fact we had not realized that Brookshire had apparently computed the spectrum based on the group-average data. As explained in the Methods section, as now clarified even better, we compute for each participant the actual spectrum of this participant’s data, and a set of surrogate spectra. We then perform a group-average of both to compute the p-value of the actual group-average based on the percentile of the distribution of surrogate averages. This send step differs from Harris & Beale, which used a one-sided t-test. The latter is most likely not appropriate in a strict statistical sense, but possibly more powerful for detecting true results compared to the percentile-based approach that we used (see l. 332ff).

(2) When results for the four experiments are reported, a reminder for the reader of how these experiments differ from each other would be useful.

We have added this in the Results section.

"considerable prevalence of differences around 4Hz, with dual‐task requirements leading to stronger rhythmicity in perceptual sensitivity". There is a striking similarity to recently published data (https://doi.org/10.1101/2024.08.10.607439) demonstrating a 4-Hz rhythm in auditory divided attention (rather than between modalities as in the present case). This could be a useful addition to the paragraph.

We have added a reference to this preprint, and additional previous work pointing in the same direction mentioned in there.

(3) There are two typos in the Introduction: "related by different from the question", and below, there is one "presented" too much.

These have been fixed.

**Reviewer #3 (Recommendations for the authors**):My major suggestion is that these results must be replicated in a new sample. I understand this is not simple to do and not always possible, but at this point, no effect is replicated from one experiment to the next, despite very small changes in protocol (especially experiment 1 vs 2). It's therefore very difficult to justify explaining the different effects as real as opposed to random effects of this particular sample. While the bootstrapping effects show the level of consistency of the effect within the sample studied, it can not be a substitute for a true replication of the results in a new sample.

We agree that only an independent replication can demonstrate the robustness of the results. We do consider experiment 1 a replication test of Ho et al. CurrBiol 2017, which results in different results than reported there. But more importantly, we consider the analysis of ‘reproducibility’ by simulating participant samples a key novelty of the present work, and want to emphasize this over the within-study replication of the same experiment. In fact, in light of the present interpretation of the data, even a within-study replication would most likely not offer a clear-cut answer.

As I said in the public review, the interpretation of the results, and of why perceptual cycles in arhythmic stimuli could be a plausible theory to begin with, is lacking. A conceptual framework would vastly improve the impact and understanding of the results.

We tried to strengthen the conceptual framework in the introduction. We believe that this is in large provided by previous work, and the aim of the present study was to explore the robustness of effects and not to suggest and discover novel effects.

Minor comments:(1) The authors adapt the difficulty as a function of performance, which seems to me a strange choice for an experiment that is analyzing the differences in performance across the experiment. Could you add a sentence to discuss the motivation for this choice?

We now mention the rationale in the Methods section and in a new section of the Results. There we also provide additional analyses on this parameter.

(2) The choice to plot the p-values as opposed to the values of the actual analysis feels ill-advised to me. It invites comparison across analyses that isn't necessarily fair. It would be more informative to plot the respective analysis outputs (spectral power, regression, or delta R2) and highlight the windows of significance and their overlap across analyses. In my opinion, this would be more fair and accurate depiction of the analyses as they are meant to be used.

We do disagree. As explained in the Methods (l. 374ff): “(Showing p-values) … allows presenting the results on a scale that can be directly compared between analysis approaches, metrics, frequencies and analyses focusing on individual ears or the combined data. Each approach has a different statistical sensitivity, and the underlying effect sizes (e.g. spectral power) vary with frequency for both the actual data and null distribution. As a result, the effect size reaching statistical significance varies with frequency, metrics and analyses.”

The fact that the level of power (or R2 or whatever metric we consider) required to reach significance differs between analyses (one ear, both ears), metrics (d-prime, bias, RT) and between analyses approaches makes showing the results difficult, as we would need a separate panel for each of those. This would multiply the number of panels required e.g. for Figure 4 by 3, making it a figure with 81 axes. Also neither the original quantities of each analysis (e.g. spectral power) nor the p-values that we show constitute a proper measure of effect size in a statistical sense. In that sense, neither of these is truly ideal for comparing between analyses, metrics etc.

We do agree thought that many readers may want to see the original quantification and thresholds for statistical significance. We now show these in an exemplary manner for the Binned analysis of Experiment 1, which provides a positive result and also is an attempt to replicate the findings by Ho et al 2017. This is shown in new Figure 5.

(3) Typo in line 555 (+ should be plus minus).(4) Typo in line 572: "Comparison of 572 blocks with minus dual task those without"(5) Typo in line 616: remove "one".(6) Line 666 refers to effects in alpha band activity, but it's unclear what the relationship is to the authors' findings, which peak around 6 Hz, lower than alpha (~10 Hz).(7) Line 688 typo, remove "amount of".

These points have been addressed.

(8) Oculomotor effect that drives greater rhythmicity at 3-4 Hz. Did the authors analyze the eye movements to see if saccades were also occurring at this rate? It would be useful to know if the 3-4 Hz effect is driven by "internal circuitry" in the auditory system or by the typical rate of eye movement.

A preliminary analysis of eye movement data was in previous Figure 8, which was removed on the recommendation of another review. This showed that the average saccade rate is about 0.01 saccade /per trial per time bin, amounting to on average less than one detected saccade per trial. Hence rhythmicity in saccades is unlikely to explain rhythmicity in behavioral data at the scale of 34Hz. We now note this in the Results.

Obleser J, Kayser C (2019) Neural Entrainment and Attentional Selection in the Listening Brain. Trends Cogn Sci 23:913-926.

Schroeder CE, Lakatos P (2009) Low-frequency neuronal oscillations as instruments of sensory selection. Trends Neurosci 32:9-18.

Schroeder CE, Lakatos P, Kajikawa Y, Partan S, Puce A (2008) Neuronal oscillations and visual amplification of speech. Trends Cogn Sci 12:106-113.

Zoefel B, Heil P (2013) Detection of Near-Threshold Sounds is Independent of EEG Phase in Common Frequency Bands. Front Psychol 4:262.